# Single-cell data integration across weakly linked modalities

Zhipeng Zhou, Yang Zhang, Zhiming Dai*

School of Computer Science and Engineering, Sun Yat-sen University, Guangzhou, Guangdong, China

* daizhim@mail.sysu.edu.cn

## Abstract

Rapid advancements in technology enables the measurement of multimodal data at single-cell resolution, but with emerging modalities that are characterized by weak correlations with other modalities. Several computational approaches attempt to integrate these weakly linked multimodal data, but face challenges regarding accurate modeling relationship between cells and learning meaningful cell representation. In this study, single-cell MultiModal data Integration through Hypergraph Contrastive Learning (MMIHCL), a deep learning-based framework that leverages an optimized adaptive k-nearest neighbor graph to model single cell pair-wise relationships for multimodal data integration is presented. MMIHCL uses hypergraph contrastive learning to capture the high-order information of a graph to produce cell representations. Comprehensive benchmarking using a multi-dimensional evaluation framework demonstrates that MMIHCL consistently delivers high-quality integration across diverse weakly linked datasets and maintains high accuracy in strongly linked scenarios. Crucially, MMIHCL exhibits versatile utility in downstream applications: it enables accurate cross-modality feature prediction via explicit cell matching, and empowers robust disease classification and drug target discovery by leveraging optimized joint embeddings. A python implementation of MMIHCL is publicly available at https://github.com/SundayChou/MMIHCL.

## Author summary

The rapid advancement of single-cell sequencing technology enables us to obtain information such as transcriptomics, epigenomics, and proteomics of individual cells. However, different omics data often "have their own stories", some of which show loose correlations. We present MMIHCL, a deep-learning framework that first lets each cell elect its most trustworthy neighbors to build a flexible network, then upgrades this network into a hypergraph where local "friend circles" contrast and learn from each other, thereby extracting more robust "cell identity cards". Beyond achieving superior technical alignment, MMIHCL demonstrates

**Data availability statement:** A python implementation of MMIHCL is publicly available at https://github.com/SundayChou/MMIHCL. This repository also provides a download link for the input data. All raw data used in this study are publicly accessible through the GEO database, Figshare, Mendeley Data, the COVID-19 Cell Atlas, and the HPAP portal. Datasets covered in this work are listed below: CITE-seq PBMC: https://atlas.fredhutch.org/data/nygc/multimodal/pbmc_multimodal.h5seurat TEA-seq PBMC: Available in the GEO database under accession code GSE158013 AB-seq BMC: https://figshare.com/articles/dataset/Expression_of_97_surface_markers_and_RNA_transcriptome_wide_in_13165_cells_from_a_healthy_young_bone_marrow_donor/13397987 CITE-seq BMC: Available in the GEO database under accession code GSE194122 CODEX tonsil: https://onlinelibrary.wiley.com/doi/10.1002/eji.202048891 and https://www.ncbi.nlm.nih.gov/geo/query/acc.cgi?acc=GSE165860 CyTOF PBMC: Available at the COVID-19 Cell Atlas (https://www.covid19cellatlas.org), "Patient donors" section. CyTOF human H1N1 & IFNG: https://data.mendeley.com/datasets/zjnpwh8m5b/1 10X-Multiome PBMC: https://www.10xgenomics.com/datasets/pbmc-from-a-healthy-donor-no-cell-sorting-10-k-1-standard-2-0-0 HPAP T1D data: Available via the HPAP portal at https://hpap.pmacs.upenn.edu/explore/donor?-by_donor Kang18 PBMC: Available in the GEO database under accession code GSE96583.

**Funding:** This work was supported by National Key Research and Development Program of China (2023YFF1204900), and Natural Science Foundation of Guangdong Province (Grant 2026A1515011206). The funders had no role in study design, data collection and analysis, decision to publish, or preparation of the manuscript.

**Competing interests:** The authors have declared that no competing interests exist.

versatile practical utility for biological research. By establishing explicit cell-to-cell matches, it allows researchers to accurately predict missing data modalities, effectively serving as a cross-omics translator. Furthermore, the high-quality cell representations empower the discovery of subtle biological signals, such as distinguishing pathological states in type 1 diabetes and identifying potential drug targets under immune stimulation. This method provides a robust tool for deciphering cellular heterogeneity and disease mechanisms in the era of multi-omics.

## Introduction

In recent years, the rapid development of single-cell sequencing technology allowed us to measure the unique properties of individual cells [1,2]. Through single-cell data, we can better understand the life state of living organisms and screen specific drugs targeting diseases [3,4]. Compared to single modality (or "omics"), exploring single-cell states from multiple modalities allows for valuable insights into the comprehensive understanding of biological systems [5,6].

Single-cell sequencing methods can simultaneously measure multimodal data within the same cell (i.e., paired data). For instance, AB-seq [7] can simultaneously measure RNA expression and cell surface protein abundance. Some semi-supervised learning-based methods, including TotalVI, scMoGNN, CLUE, and MultiVI [8–11], have been developed to integrate paired single-cell data. However, no technology can measure all modalities within an individual cell, and most of single-cell multimodal data are unpaired.

A key factor influencing the integration of single-cell multimodal data is the strength of linkage between modalities. As defined in a previous study [12]: A feature is "linked" between two modalities if it was measured in, or can be predicted by both modalities. These linked features can facilitate alignment of feature spaces of multi-modalities [13]. Two modalities are defined to show "strong linkage" if there is a great number of linked features showing strong correlations between these two modalities. For example, between single-cell RNA sequence (scRNA-seq) and single-cell Assay for Transpose-Accessible Chromatin sequence (scATAC-seq) data, RNA level of each gene is strongly "linked" to chromatin accessibility in gene promoter regions [14,15], and these two modal data are available for most genes. Most existing multimodal data integration methods are designed for scenarios of "strong linkage" above. Traditional unsupervised learning-based strong linkage integration methods include Seurat V3, Liger, and Harmony [5,16,17]. Subsequent strong linkage-based methods, including UniPort and MARIO [18,19], etc., have a variety of innovations in architectural design.

Some modalities have a small number of linked features, and/or the linked features show weak cross-modality correlation, in this situation we refer to as "weak linkage". For example, a single dataset of protein-based assays is only for a small number of proteins, with limited linked features with single-cell sequencing (e.g.

scRNA-seq) datasets. These weakly linked features pose a challenge to produce high-quality pairwise cell-type matching. As spatial proteomic technologies become widespread [20,21], several methods such as scConfluence, MaxFuse, and the recently developed CelLink [12,22,23], have recently been developed for cross-modality integration in scenarios of weak linkage. Specifically, CelLink addresses the issues of weak feature correlation and imbalanced cell populations through an iterative unbalanced optimal transport framework. Furthermore, scMRDR [24] further enhances the scalability and flexibility of unpaired integration across diverse modalities by learning regularized disentangled representations. These methods perform better than strong linkage-based methods on weak linkage data.

However, there are still some challenges in data integration across weakly linked modalities. First, single-cell data with different modalities have their own feature space, especially the feature number of proteomic data is extremely limited [25]. The limited prior knowledge between proteomics and other omics data is an obstacle for accurate matching of these weakly linked modality data. Second, the distribution varies with intra-modal cell types, e.g., the cell numbers of different cell types are generally unbalanced [26]. Finally, how to model the complex relationship between cells to improve cell representation learning remains to be solved.

To address these challenges, we propose a method for single-cell MultiModal data Integration through Hypergraph Contrastive Learning (MMIHCL). MMIHCL uses an adaptive k-Nearest Neighbor (akNN) graph for the imbalance of cell numbers, and uses hypergraph contrastive learning to learn the high-order relationships between cells. An iterative strategy is introduced to refine cell representation learning and cross-modal cell-type matching based on limited linked features. Crucially, our framework is designed to go beyond simple alignment: it explicitly matches cells to enable cross-modality feature prediction and learns optimized joint embeddings that empower robust downstream biological discovery. Our main contributions are summarized as follows:

- We propose MMIHCL, a novel deep learning framework that integrates an akNN graph with hypergraph contrastive learning to effectively address data imbalance problem and model high-order cellular relationships.

- We demonstrate that MMIHCL significantly outperforms state-of-the-art (SOTA) methods across diverse weakly linked datasets while maintaining high accuracy in strongly linked scenarios. Furthermore, extensive evaluations—including ablation study, parameter sensitivity analysis, and scalability tests on large-scale datasets (100k cells)—validate the robustness and computational efficiency of our framework.

- We showcase the versatile utility of MMIHCL in downstream biological applications. Specifically, MMIHCL enables precise cross-modality feature prediction via explicit cell matching and empowers robust disease classification and drug target discovery by leveraging optimized joint embeddings.

## Results

### Overview of MMIHCL

The MMIHCL pipeline consists of three main stages, and its overview pipeline is shown in Fig 1. In stage 1, the adjacent cell graph of each modality is constructed based on akNN. Considering the scenarios of unbalanced cell numbers of different cells, akNN allows the optimized number of neighbors to be adaptively chosen for each cell, compared to the number to be constant $k$ in traditional kNN. In stage 2, cross-modal linked features are used to generate an initial many-to-many cell matching by a hypergraph embedding learning module. Compared with general graph, hyperedges in hypergraph are able to connect with more than two vertices. Therefore, message passing is extended to multiple sources and targets on hypergraphs, which efficiently encodes high-order relations between vertices (i.e., cells). In stage 3, all features from each modality and the hypergraph module are used to iteratively optimize cell embeddings and generate the final cell matching and co-embeddings. For more details on MMIHCL's pipeline, please refer to Materials and methods.

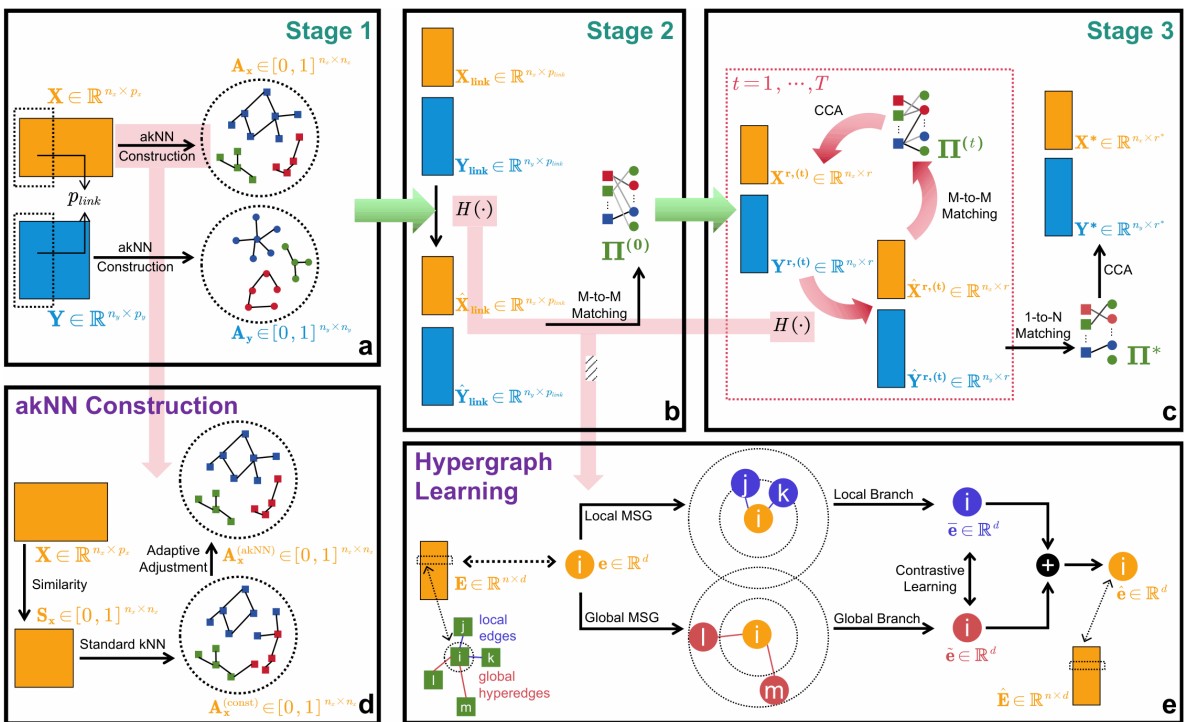

**Fig 1. Overview pipeline of MMIHCL. (a)** The input modality matrices **X** and **Y** share $p_{link}$ linked features. Weighted adjacency cell graphs **A_x** and **A_y** are constructed through the akNN module. **(b)** Linked features are embedded via the hypergraph operator $H(\cdot)$ to generate initial embeddings, followed by a Many-to-Many (M-to-M) matching process to obtain the initial matching $\Pi^{(0)}$. **(c)** The workflow iteratively updates cell embeddings and matching for $t = 1, 2, \cdots, T$ iterations. Each loop involves $H(\cdot)$, Canonical Correlation Analysis (CCA) [27], and M-to-M matching. The final outputs are the optimized joint embeddings **X***, **Y*** and the 1-to-N matching $\Pi^*$. **(d)** Feature similarity **S_x** is calculated to transition from a constant $k$ in standard kNN to a cell-specific adaptive $k_i$, yielding the final graph $\mathbf{A_x^{(akNN)}}$. **(e)** For a target cell $i$, local and global message-passing branches generate embeddings $\bar{\mathbf{e}}$ and $\tilde{\mathbf{e}}$, respectively. Contrastive learning is employed between the two branches to enhance representation robustness, followed by a fusion step to obtain the final learned embedding $\hat{\mathbf{e}}$.

## MMIHCL shows superiority on the weak linkage dataset integration

We categorize the linkage intensity between two modalities based on the linkage ratio $\rho$ defined in Eq (1). Specifically, two datasets are defined as weakly linked if $\rho < 10\%$. We used seven methods published for multimodal integration as baselines, including traditional method Seurat [16], two strong linkage-based methods MARIO [19] and UniPort [18], and four weak linkage-based methods MaxFuse [12], scConfluence [22], CelLink [23], and scMRDR [24]. For detailed implementation of each method, please refer to Section 1 in S1 File. To test the integration performance of MMIHCL and baselines on weakly linked datasets, we conducted experiments on datasets generated by different sequencing technologies, including CITE-seq PBMC [28], TEA-seq PBMC [29], AB-seq BMC [7], CITE-seq BMC [28], and CODEX tonsil [30,31], which were used in the study of MARIO [19], MaxFuse [12], scConfluence [22], and CelLink [23]. Considering randomness in the algorithm, each method was run five times on each dataset. Unless otherwise specified, this protocol of five independent repetitions applies to all quantitative analyses presented in this study.

The quantitative integration performance on weak linkage datasets is summarized in Fig 2. To avoid the bias of single metrics, we employed the weighted overall score ($S_{overall}$) to evaluate the methods holistically. As shown in Fig 2a, MMIHCL consistently delivers high-quality integration across all five datasets. The mean $S_{overall}$ of MMIHCL consistently surpasses the distributions of all baseline methods, demonstrating a statistically significant advantage (Wilcoxon signed-rank tests,

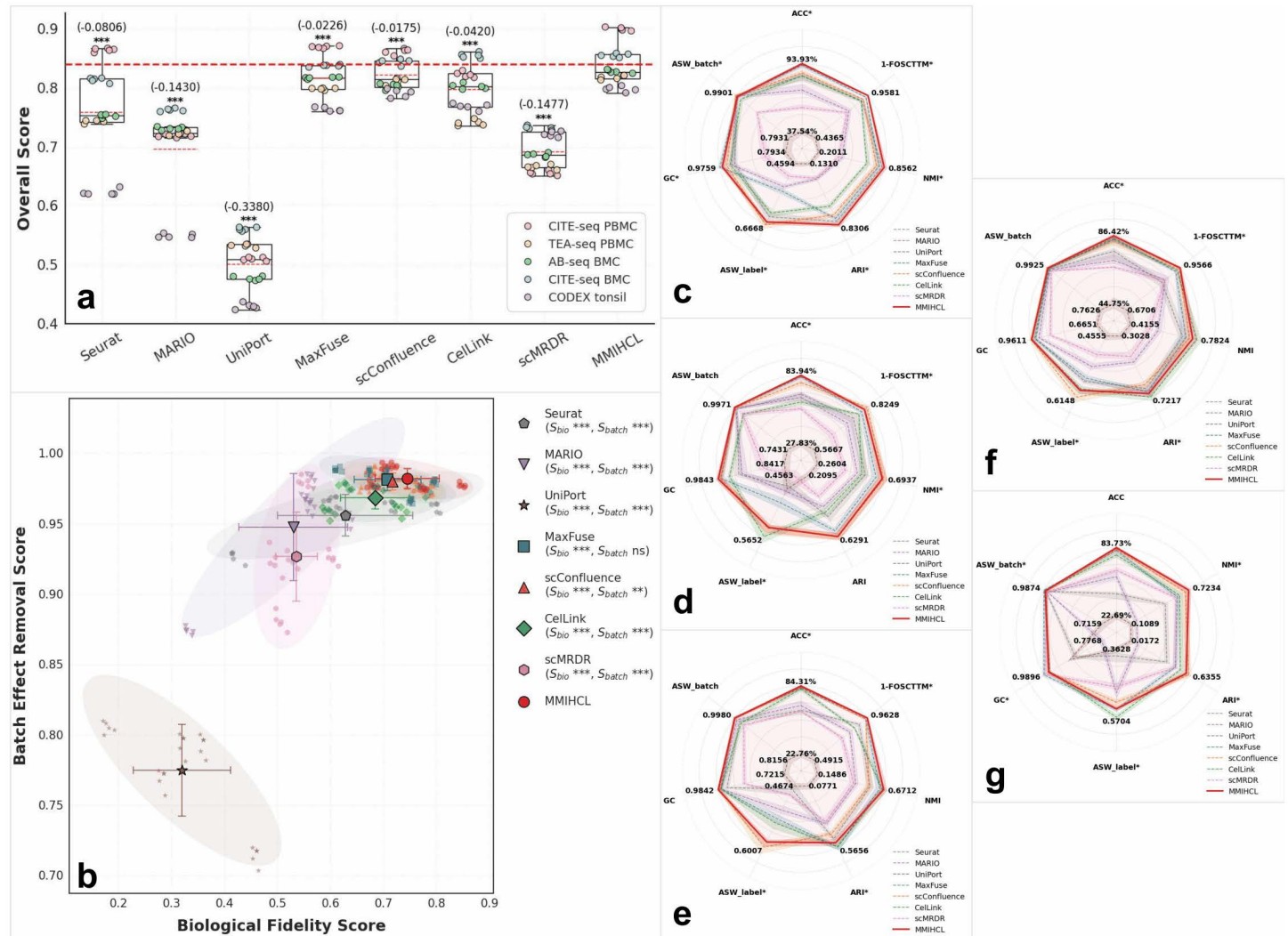

**Fig 2. Comprehensive performance evaluation on weak linkage datasets. (a)** Boxplots show the weighted overall scores balancing biological fidelity and multimodal alignment. The horizontal bold red dashed line represents the MMIHCL mean baseline. Numerical values in parentheses denote mean differences relative to MMIHCL. The boxplots are defined as follows: the minimum is calculated as the 25th percentile minus 1.5 times the Inter-Quartile Range (IQR), and the maximum is calculated as the 75th percentile plus 1.5 times the IQR. The hinges of the box represent the IQR, while the whiskers extend to 1.5 times the IQR. The black line and the red dotted line in the box plot represent the median and the mean respectively, and the bounds of the box correspond to the 25th and 75th percentiles. Statistical significance for (a) and (b) was determined by two-sided Wilcoxon signed-rank tests (***: $P < 0.001$, **: $P < 0.01$, *: $P < 0.05$, ns: not significant). **(b)** Scatter plot illustrates the balance between biological fidelity ($S_{bio}$) and batch effect removal ($S_{batch}$). Diamonds, error bars, and shaded 95% confidence ellipses represent centroids, standard deviations, and the distribution of results across five repetitions, respectively. Statistical significance symbols in the legend also refer to MMIHCL as the reference. **(c-g)** Radar charts display performance across seven metrics (ACC, 1-FOSCTTM, NMI, ARI, $ASW_{label}$, GC, and $ASW_{batch}$) for: **(c)** CITE-seq PBMC, **(d)** TEA-seq PBMC, **(e)** AB-seq BMC, **(f)** CITE-seq BMC, and **(g)** CODEX tonsil. Significance markers for metrics indicate statistical differences between the two top-ranked methods (determined by two-sided Wilcoxon signed-rank tests). NOTE: Unless otherwise specified, definitions for plot elements and significance testing remain consistent for similar figures throughout this study.

$P < 0.001$). Specifically, MMIHCL shows a substantial lead over traditional methods, outperforming Seurat and MARIO with mean differences of 0.0806 and 0.1430, respectively. While the recently introduced scMRDR achieves a moderate overall score, it still lags behind MMIHCL by a notable margin ($\Delta = -0.1477$), indicating that pure generative alignment

struggles with severe cross-modality feature sparsity. Even compared to SOTA weak linkage integrators, such as Max-Fuse ($\Delta$ = –0.0226), scConfluence ($\Delta$ = –0.0175), and CelLink ($\Delta$ = –0.0420), MMIHCL maintains a significant lead.

Fig 2b illustrates the optimal trade-off achieved by MMIHCL between biological fidelity ($S_{bio}$) and batch effect removal ($S_{batch}$). MMIHCL uniquely occupies the top-right quadrant with centroid coordinates at ($S_{bio}$ = 0.7449, $S_{batch}$ = 0.9822), significantly surpassing the runner-up method, scConfluence, which sits at ($S_{bio}$ = 0.7172, $S_{batch}$ = 0.9799). The compact 95% confidence ellipses further highlight MMIHCL's exceptional robustness (area = 0.0052) compared to the high variance observed in Seurat (area = 0.0166) or UniPort (area = 0.0260).

This superiority is further decomposed in the multi-dimensional profiles shown in Fig 2c–2g. For instance, on the CITE-seq PBMC dataset (Fig 2c), MMIHCL dominates the performance chart, achieving an ACC of 93.93% and an NMI of 0.8562, which are 1.11% and 0.0221 higher than those of the second-best method (MaxFuse and Seurat), respectively. Similarly, in the challenging CODEX tonsil dataset (Fig 2g), MMIHCL maintains high metric consistency with a GC score of 0.9728, effectively preserving local neighborhood structures where other methods struggle. Collectively, MMIHCL encompasses the largest area in all five radar charts (mean normalized area = 2.5787, surpassing scConfluence by 6.23%), confirming its superior versatility across diverse sequencing technologies.

We also use Uniform Manifold Approximation and Projection (UMAP) [32] to visualize joint embedding across the two modalities. Here we only show the UMAP results of all methods on CITE-seq PBMC in Fig 3. Limited by space, for results that include all methods on other four weak linkage datasets, please refer to S1–S4 Figs. In UMAP images colored by data modality, MMIHCL always integrates multimodal data in the joint embedding space well, showing multimodal alignment. In contrast, the latent space of UniPort is almost completely unable to bridge the gaps between RNA and protein data. Furthermore, while advanced baselines such as MaxFuse, scConfluence, CelLink, and scMRDR show improved integration performance, MMIHCL consistently exhibits the most distinct cell-type boundaries and preservation of biological heterogeneity. As another contrast, in MARIO's feature space, sometimes the various types (mainly CD4 T, CD8 T, DC, and NK) of cells are mixed together in a chaotic manner, making it difficult to distinguish.

## MMIHCL shows competitive performance on the strong linkage dataset integration

We also tested whether MMIHCL shows competitive performance on strongly linked modalities, which, according to the criteria in Eq (1), are defined as scenarios where $\rho \geq 10\%$. We benchmarked MMIHCL against six baseline methods — Seurat [16], MARIO [19], MaxFuse [12], scConfluence [22], CelLink [23], and scMRDR [24] — excluding UniPort from this analysis due to runtime collapse. We used datasets of strongly linked modalities, including CITE-seq & CyTOF PBMC [28,33], CyTOF human H1N1 & IFNG [34,35], and 10X-Multiome PBMC [36], which were used in the study of MARIO [19] and MaxFuse [12].

The quantitative integration performance on strong linkage datasets is summarized in Fig 4. In addition, we show the full UMAP results on other two strong linkage datasets in S5 Fig and S6 Fig. In these high-correlation scenarios, baseline methods generally perform well due to the abundance of shared signals. However, as shown in Fig 4a, MMIHCL still maintains the highest performance ceiling. The mean $S_{overall}$ of MMIHCL consistently surpasses competing methods, achieving a statistically significant lead over Seurat ($\Delta$ = –0.0509) and MARIO ($\Delta$ = –0.1199). Even against robust deep learning baselines, MMIHCL retains a consistent advantage, as evidenced by the significance markers (***, $P < 0.001$) over Max-Fuse ($\Delta$ = –0.0612) and CelLink($\Delta$ = –0.0378).

Fig 4b further reveals the stability of MMIHCL in high-fidelity integration. The method's centroid resides in the extreme top-right corner ($S_{bio}$ = 0.8112, $S_{batch}$ = 0.9908), indicating an optimal balance between biological conservation and batch effect removal. Notably, MMIHCL exhibits significantly tighter 95% confidence ellipses (area = 0.0012) compared to the high variance observed in MaxFuse (area = 0.0025), CelLink (area = 0.0028), and scMRDR (area = 0.0074), demonstrating superior reproducibility across independent runs.

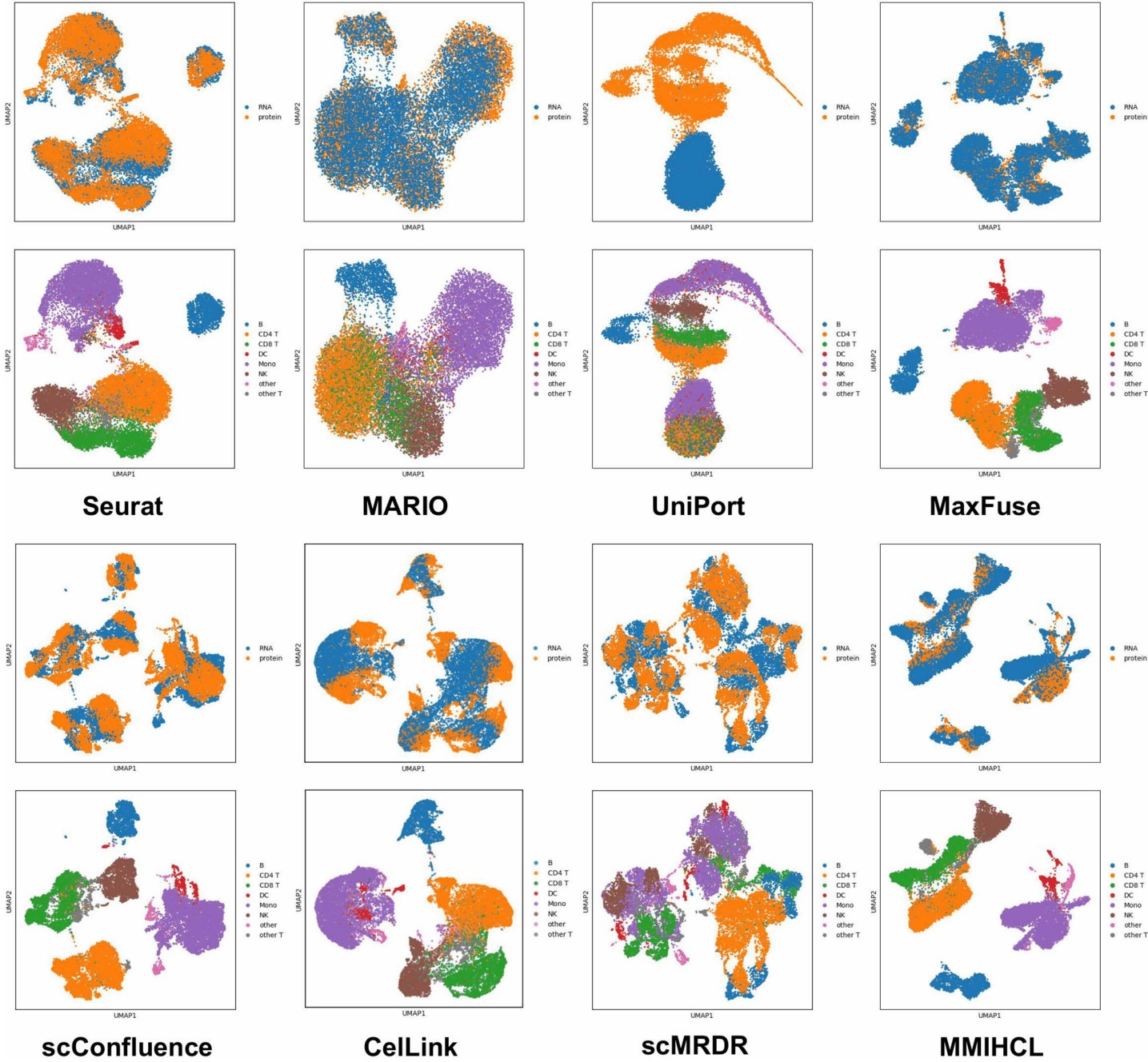

**Fig 3. UMAP visualization on CITE-seq PBMC dataset.** The first and third row subgraphs are colored by data modality, and the second and fourth row subgraphs are colored by cell type. Other UMAP graphs are also arranged in this way.

Detailed metrics in Fig 4c–4e confirm this trend across specific technologies. For instance, in the CyTOF human H1N1 & IFNG dataset (Fig 4d), MMIHCL achieves a NMI of 0.9417 and an ARI of 0.9285, effectively distinguishing immune cell subtypes where other methods show minor fluctuations. Collectively, MMIHCL encompasses the most comprehensive area in the radar charts (mean normalized area = 2.2836), significantly outperforming the runner-up method, scConfluence

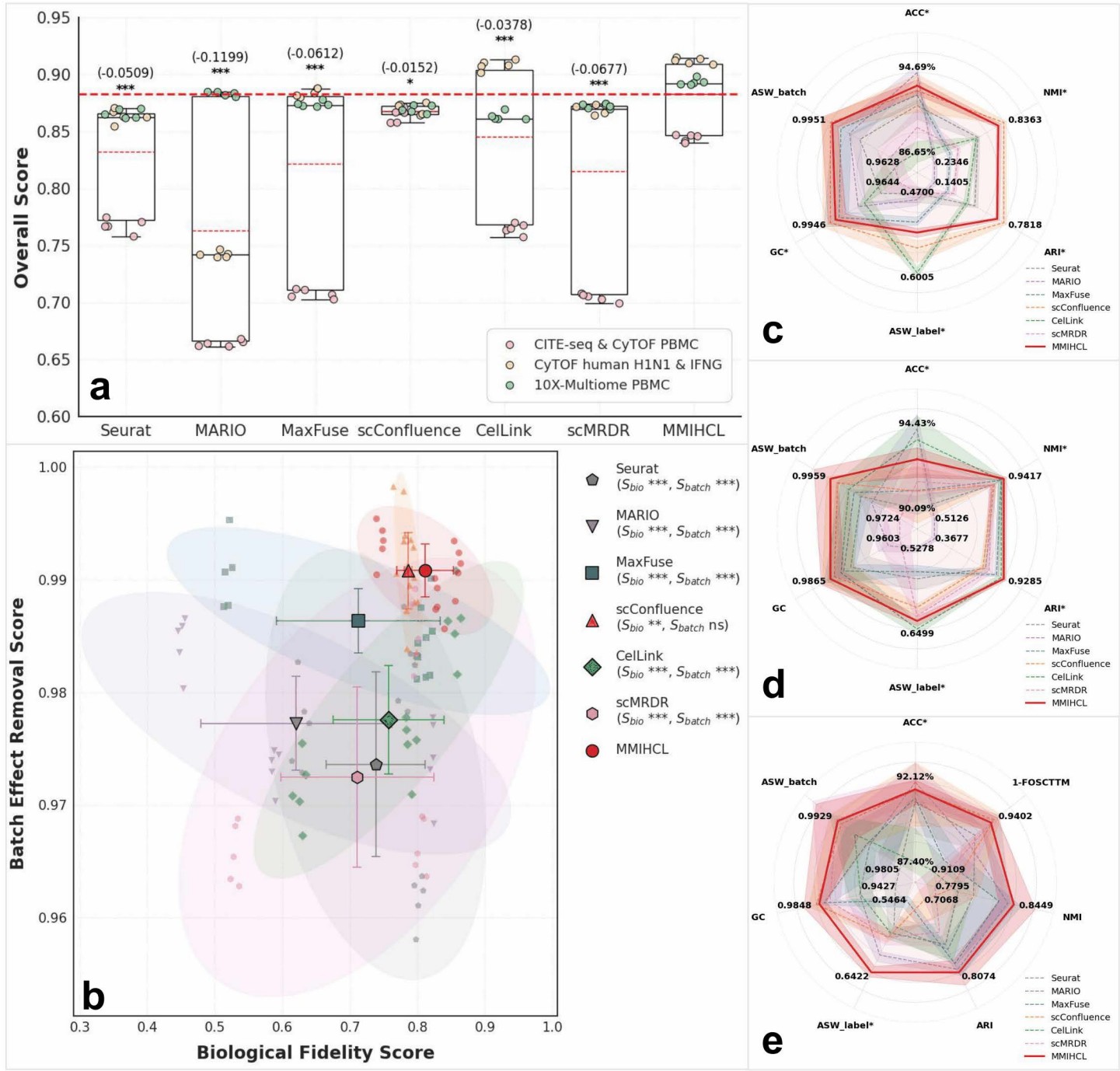

**Fig 4. Comprehensive performance evaluation on strong linkage datasets. (a)** Boxplots show the weighted overall scores balancing biological fidelity and multimodal alignment. **(b)** Scatter plot illustrates the balance between biological fidelity ($S_{bio}$) and batch effect removal ($S_{batch}$). **(c-e)** Radar charts display performance across seven metrics (ACC, 1-FOSCTTM, NMI, ARI, $ASW_{label}$, GC, and $ASW_{batch}$) for: **(c)** CITE-seq & CyTOF PBMC, **(d)** CyTOF human H1N1 & IFNG, and **(e)** 10X-Multiome PBMC.

(area = 1.7286), and exhibiting nearly double the coverage of other advanced baselines like CelLink (area = 1.1401) and scMRDR (area = 0.9963). This geometric dominance confirms that MMIHCL remains a top-tier performer even in scenarios where baseline methods are highly competitive.

The UMAPs shown in Fig 5 indicate that the performance of MMIHCL is comparable to that of baselines. Visually, MMIHCL demonstrates superior eliminating multimodal gap and biological heterogeneity protection capabilities compared to other methods. As a comparison, in the UMAP result of scConfluence about data type, some H1N1 cells on the left do

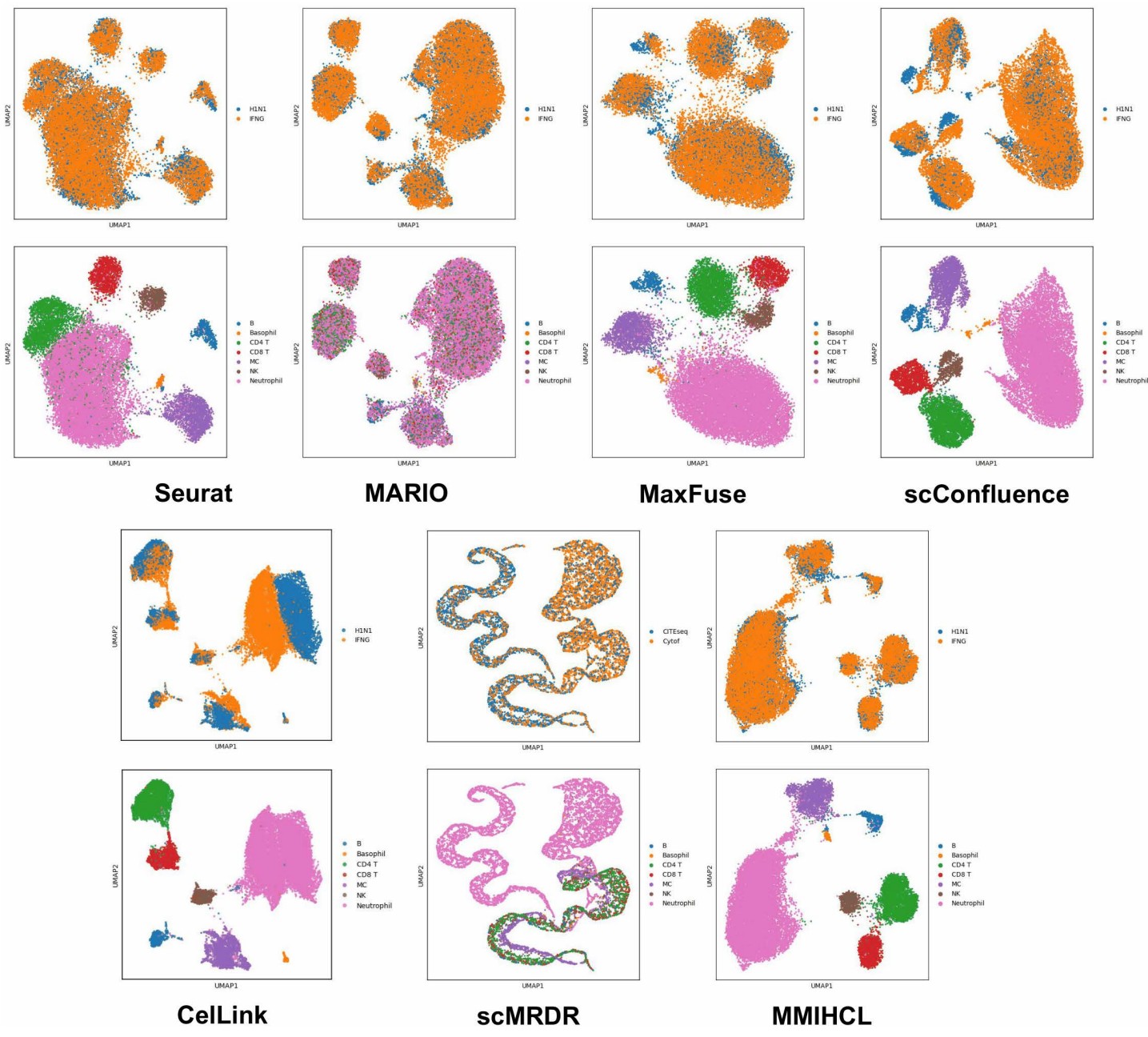

**Fig 5. UMAP visualization on CyTOF human H1N1 & IFNG dataset.**

not fuse sufficiently with IFNG cells. Besides, we can see that MARIO's joint embeddings do not clearly separate different types of cells, especially Neutrophils are mixed together with many other types of cells.

## MMIHCL enables accurate cross-modality feature prediction via explicit cell matching

Cross-modality feature prediction serves as a rigorous benchmark for alignment fidelity. We benchmarked MMIHCL against other tools representing distinct strategies. Specifically, MMIHCL executes feature prediction through a "regression-free" mechanism: it utilizes the learned explicit cell-cell matching matrix to directly project the high-dimensional feature profiles from the reference modality onto the query cells. For detailed implementation protocols regarding how each baseline method specifically performs cross-modality feature prediction, please refer to Section 1 in S1 File. By leveraging this explicit matching paradigm, MMIHCL relies on identifying biologically true correspondences rather than merely aligning statistical distributions, thereby ensuring fidelity in recovering quantitative biological signals.

We first evaluated this framework on the CITE-seq PBMC [28] dataset to predict surface protein abundance from gene expression level. MMIHCL achieved the highest cell-wise Pearson Correlation Coefficient (PCC) level (as shown in Fig 6a), attaining a median of 0.9263 compared to 0.9208 for the runner-up MaxFuse [12]. Unlike baselines such as MARIO [19], which exhibited broad variances, MMIHCL showed a compact distribution concentrated at the upper range. These improvements were statistically significant against all baselines (two-sided Wilcoxon signed-rank tests, $P < 0.001$). Furthermore, Cumulative Distribution Function (CDF) analysis in Fig 6b demonstrated that MMIHCL yielded the largest Area Under Curve (AUC), 0.8775. Specifically, 80.27% of cells predicted by MMIHCL surpassed a stringent threshold of $r > 0.8$, outpacing the closest competitors Maxfuse (77.44%) and CelLink [23] (69.24%) by a substantial margin. These results underscore the substantial advantage of explicit cell matching in recovering quantitative protein signals.

We further assessed biological fidelity via side-by-side heatmaps in Fig 6c. MMIHCL exhibited exceptional textural resemblance to the ground truth, effectively preserving intra-population heterogeneity. For instance, in predicting CD11c on Dendritic Cell (DC), MMIHCL accurately reproduced the diverse expression gradients. In contrast, MaxFuse showed a systematic underestimation of signal intensity, rendering the DC cluster noticeably dimmer and failing to reach the peak expression levels observed in the biological baseline. This precision was corroborated by UMAP visualizations (Fig 6d) of lineage markers. MMIHCL faithfully recapitulated the spatial distribution of CD3.1 (T cells) and CD19 (B cells), recovering high-intensity signals with sharp boundaries indistinguishable from the ground truth.

To validate robustness beyond the central dogma, we extended our evaluation to the 10X-Multiome PBMC [36] dataset in S7 Fig, predicting gene expression from sparse chromatin accessibility data. MMIHCL maintained high predictive concordance with a median PCC of 0.7197 (S7A Fig), showing no statistically significant difference from MARIO (0.7210, $P \geq 0.05$) while significantly outperforming scConfluence [22] (0.7158, $P < 0.05$). Regarding CDF (S7 b Fig), MMIHCL achieved a competitive AUC of 0.7061, ranking a close second to the top-performing MARIO (AUC = 0.7079) with a negligible margin. Qualitatively, we highlighted *SLC4A10*, a canonical transcriptomic signature of Mucosal-Associated Invariant T (MAIT) cells. While MMIHCL consistently recovered its expression, MARIO exhibited a distinct signal dropout—visible as a contradictory dark void—in a subset of the cluster (S7c Fig). Furthermore, UMAP visualizations (S7d Fig) confirmed MMIHCL's ability to faithfully reconstruct the spatial patterns of lineage markers like *MS4A1* and *CST3*, preserving the biological manifold's topology.

## MMIHCL empowers robust disease classification and drug target discovery

Beyond the explicit cell matching that facilitates the cross-modality feature prediction described above, MMIHCL can also utilize the optimized joint embedding to reveal deeper biological heterogeneity. Here, we utilized the unpaired HPAP [37] and Kang18 PBMC [38] dataset to investigate the potential of MMIHCL in downstream biological discovery.

We first evaluated the quality of the learned embeddings on the HPAP dataset, which comprises RNA and protein modalities from donors with Type 1 Diabetes (T1D) and healthy controls. As visualized in the top row of Fig 7a, MMIHCL achieved

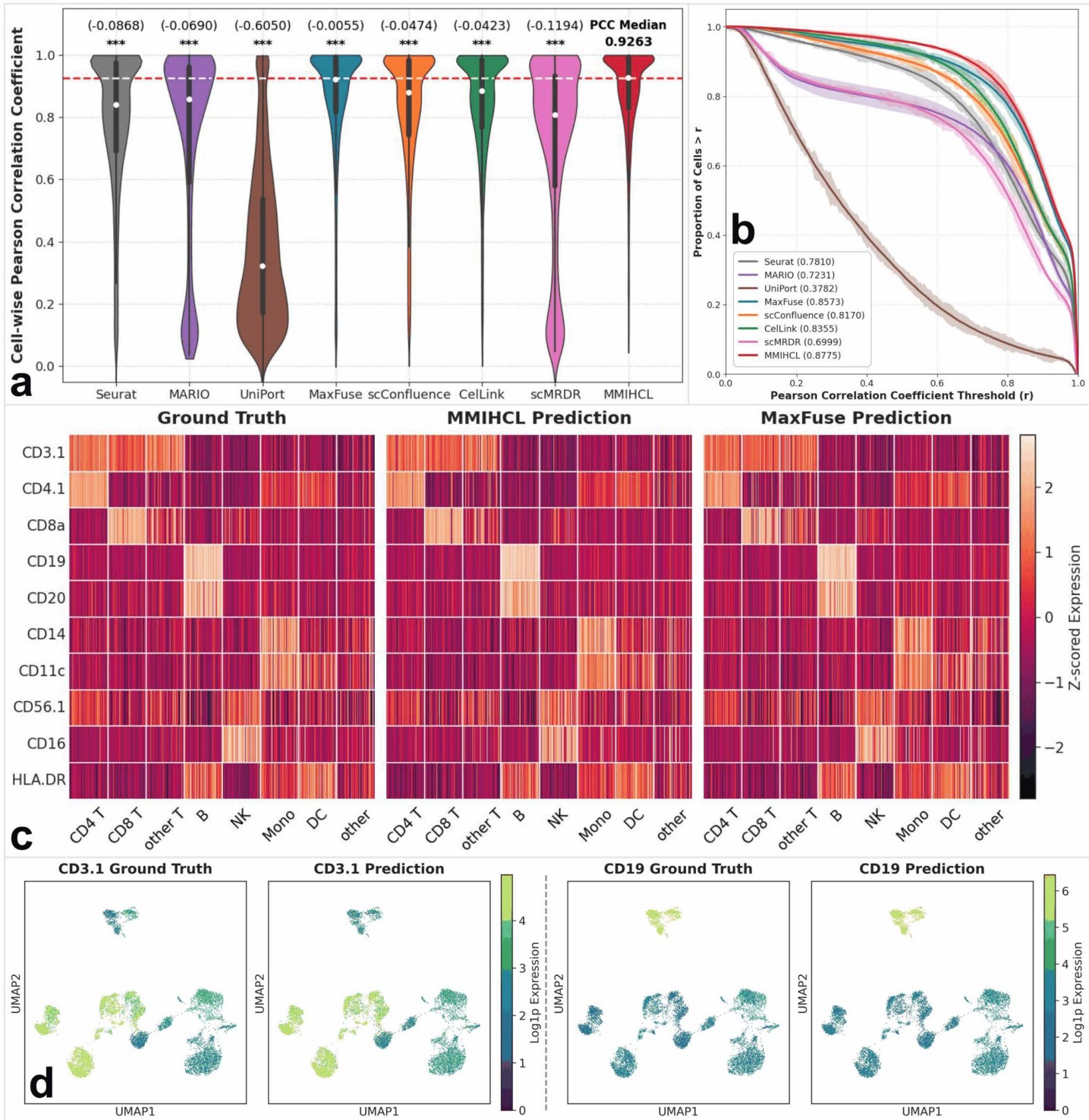

**Fig 6. Performance evaluation of cross-modality feature prediction on the CITE-seq PBMC dataset. (a)** Violin plots displaying the distribution of cell-wise PCCs between ground truth and predicted surface protein abundances. The white dot represents the median PCC, and the thick bar indicates the interquartile range. The numbers above the violins indicate the difference in median PCC relative to MMIHCL. Statistical significance was determined using two-sided Wilcoxon signed-rank tests (***: $P<0.001$, **: $P<0.01$, *: $P<0.05$, ns: not significant). **(b)** CDF curves illustrating the proportion of cells

(y-axis) surpassing specific PCC thresholds (x-axis). The translucent shading surrounding each curve represents the standard deviation, and the values in parentheses within the legend denote the AUC for each method. **(c)** Side-by-side heatmaps comparing the z-scored expression of 10 representative surface proteins (e.g., CD3.1, CD19, CD14) across annotated cell types for ground truth, MMIHCL prediction, and MaxFuse prediction. Rows represent protein markers, and columns represent individual cells sorted by cell type. **(d)** UMAP visualizations of ground truth versus predicted expression for two lineage-specific protein markers: CD3.1 (T cells) and CD19 (B cells).

the most distinct separation between disease states compared to the unintegrated Principal Component Analysis (PCA) [39] baseline and integration methods including Seurat [16] and MaxFuse [12]. Quantitative metrics confirmed this observation, with MMIHCL yielding the highest status NMI (0.4618) and ARI (0.5439), indicating its capability to capture robust pathological signatures. Crucially, while effectively distinguishing disease conditions, MMIHCL also preserved fine-grained biological identity within the disease state. In the T1D subpopulation (Fig 7a, bottom row), while MaxFuse yielded a marginally higher NMI (0.3701) compared to MMIHCL (0.3559), MMIHCL achieved a superior ARI (0.4632). This higher ARI suggests that MMIHCL more accurately retains the structural integrity of Alpha, Delta-PP, and Ductal cell types without merging distinct clusters, thereby maintaining the intrinsic cellular heterogeneity essential for precise disease subtyping.

To further demonstrate MMIHCL's utility in drug target discovery, we analyzed the Kang18 PBMC dataset (PBMCs stimulated with interferon-beta, IFN-$\beta$) using a neighborhood aggregation strategy to mitigate technical noise. By smoothing gene expression via kNN graphs constructed from the joint embeddings, MMIHCL successfully discovered subtle drug-response signals. For representative interferon-stimulated genes including *CXCL10*, *CXCL11*, and *CCL2*, MMIHCL-aggregated profiles exhibited the sharpest separation between control and simulated groups (Fig 7b). Specifically for *CXCL10*, MMIHCL achieved the most significant statistical difference (Welch's t-tests, $p = 1.10 \times 10^{-120}$) compared to Seurat ($p = 6.82 \times 10^{-119}$) and MARIO [19] ($p = 2.60 \times 10^{-112}$). This superior signal-to-noise ratio is critical for minimizing false negatives in screening potential therapeutic targets. Extending this to a genome-wide scale, the volcano plot (Fig 7c) revealed that MMIHCL identifies Differentially Expressed Genes (DEGs) with a remarkably broad dynamic range (Log2FC spanning from -6.28 to +5.14). This wide spectrum allows for the simultaneous detection of both drastically altered effectors (e.g., *LIMK2* and *CEBPD*) and subtly regulated but biologically significant markers (e.g., *ANO9* and *CXCL10*), thereby providing a comprehensive and high-confidence candidate list for mechanism-of-action studies.

## Ablation study

To comprehensively validate the contribution of each component in MMIHCL, we compared the full model against eight variant configurations (summarized in Table 1):

1. **w/o Filter**: This setup removes the reliability filtering mechanism during matching pair generation. It bypasses the quality control threshold, retaining all identified matching pairs—regardless of their confidence levels—for the subsequent joint embedding learning.

2. **w/o akNN**: This setup replaces the adaptive neighbor selection with a fixed number of neighbors (constant $k$) for all cells. It disregards the local density variations captured by the akNN module, forcing a uniform receptive field across both sparse and dense cell populations.

3. **w/o Hypergraph with CL**: In this setup, the hypergraph branch is removed. Consequently, the Contrastive Learning (CL) scheme is modified to align the raw initial embeddings with the GCN-aggregated embeddings (from the normal graph view), rather than aligning two distinct graph views (normal & hypergraph).

4. **w/o Hypergraph w/o CL**: This setup further degrades the previous setup by removing the contrastive objective entirely. It simply performs a direct summation of the raw initial embeddings and the GCN-aggregated embeddings, feeding the combined result into subsequent steps without any self-supervised alignment.

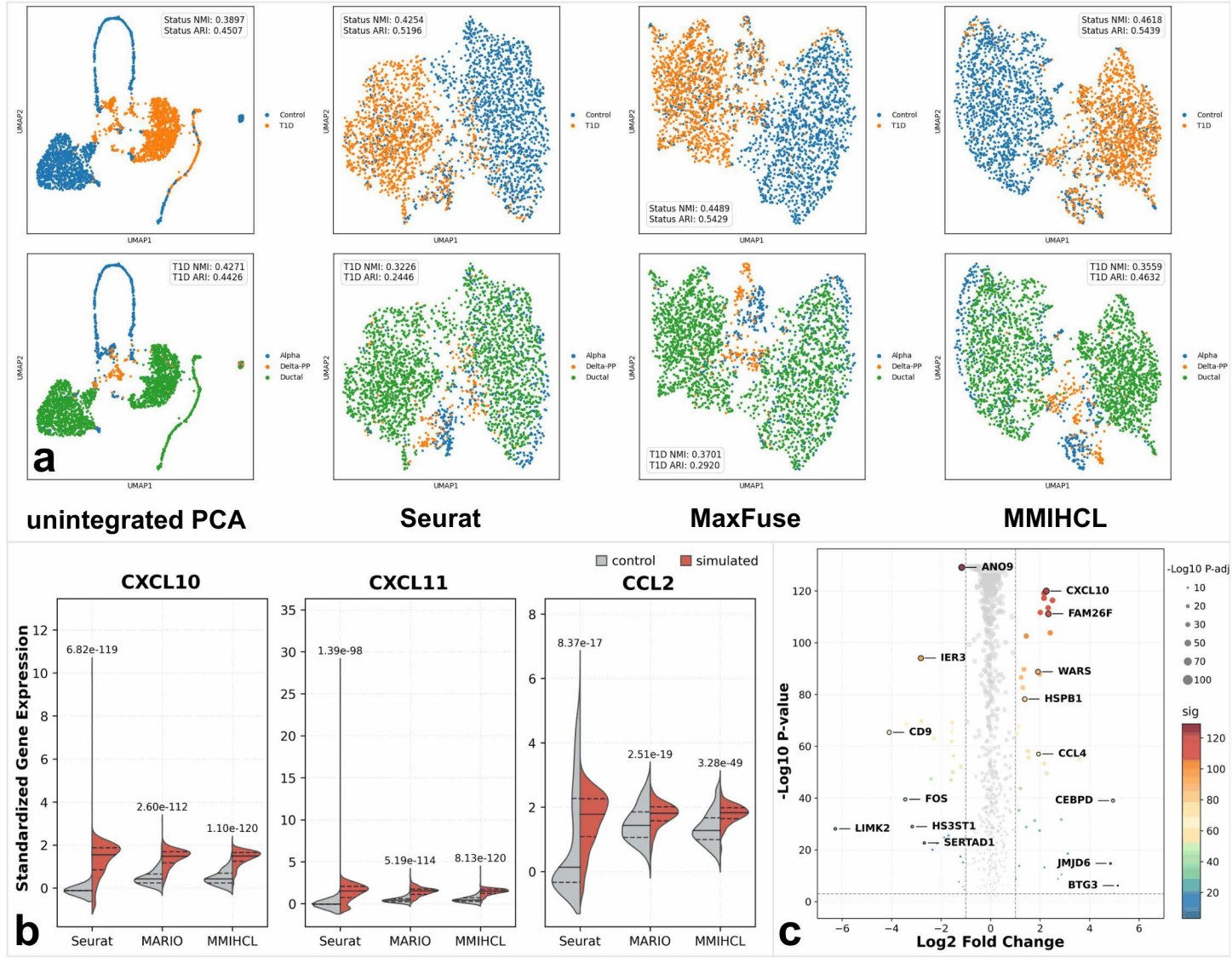

**Fig 7. Application of MMIHCL in disease classification and drug target discovery. (a)** UMAP visualization of joint embeddings on the HPAP dataset. The top row subgraphs are colored by disease status (Control vs. T1D), and the bottom row subgraphs are colored by cell type. The clustering performance metrics (NMI and ARI) in the top row are calculated using all cells, whereas those in the bottom row are computed using exclusively the T1D subpopulation. **(b)** Split violin plots comparing the expression distributions of three representative interferon-stimulated genes (*CXCL10*, *CXCL11*, and *CCL2*) from the Kang18 PBMC dataset across Seurat, MARIO, and MMIHCL. The numeric values annotated above the violins indicate the statistical significance (P-values) derived from the Welch's t-tests. **(c)** Volcano plot visualizing the DEGs identified by MMIHCL on the Kang18 PBMC dataset. Significantly up-regulated and down-regulated genes are highlighted in color, whereas non-significant genes are displayed in gray.

5. **w/o Filter & akNN**: This setup simultaneously removes the reliability filtering mechanism and the adaptive neighbor selection. It forces the model to construct a rigid intra-modality graph topology while concurrently feeding unpurified cross-modality matching pairs into the joint learning phase.

6. **w/o Filter & Hypergraph CL**: This setup disables both the reliability filter and the hypergraph branch, which inherently removes the contrastive learning objective entirely. It relies solely on the sum of the original embeddings and

**Table 1. Ablation study of MMIHCL under weak and strong linkage scenarios.**

| Scenario → | Weak Linkage | | | | Strong Linkage | | | |
|---|---|---|---|---|---|---|---|---|
| Metric → | ACC | $S_{bio}$ | $S_{batch}$ | $S_{overall}$ | ACC | $S_{bio}$ | $S_{batch}$ | $S_{overall}$ |
| Setup ↓ | | | | | | | | |
| Full MMIHCL | 86.47% | 0.7449 | 0.9822 | 0.8398 | 92.61% | 0.8112 | 0.9908 | 0.8831 |
| w/o Filter | 84.85%*** (-1.62%) | 0.7264*** (-0.0185) | 0.9786*** (-0.0036) | 0.8273*** (-0.0125) | 92.25%** (-0.36%) | 0.8004*** (-0.0108) | 0.9881*** (-0.0027) | 0.8755*** (-0.0076) |
| w/o akNN | 82.84%*** (-3.63%) | 0.7022*** (-0.0427) | 0.9731*** (-0.0091) | 0.8106*** (-0.0261) | 92.31%* (-0.30%) | 0.7911*** (-0.0201) | 0.9853*** (-0.0055) | 0.8688*** (-0.0143) |
| w/o Hyper-graph with CL | 85.98%ns (-0.49%) | 0.7397*** (-0.0052) | 0.9832ns (+0.0010) | 0.8371*** (-0.0027) | 92.51%ns (-0.10%) | 0.8079** (-0.0033) | 0.9917ns (+0.0009) | 0.8815** (-0.0016) |
| w/o Hyper-graph w/o CL | 82.90%*** (-3.57%) | 0.7021*** (-0.0428) | 0.9733*** (-0.0089) | 0.8106*** (-0.0292) | 92.31%** (-0.30%) | 0.7905*** (-0.0207) | 0.9851*** (-0.0057) | 0.8684*** (-0.0147) |
| w/o Filter & akNN | 82.01%*** (-4.46%) | 0.6965*** (-0.0484) | 0.9719*** (-0.0103) | 0.8063*** (-0.0335) | 92.13%** (-0.48%) | 0.7881*** (-0.0231) | 0.9842*** (-0.0066) | 0.8665*** (-0.0166) |
| w/o Filter & Hypergraph CL | 80.84%*** (-5.63%) | 0.6877*** (-0.0572) | 0.9703*** (-0.0119) | 0.8008*** (-0.0390 | 91.97%** (-0.64%) | 0.7841*** (-0.0271) | 0.9829*** (-0.0079) | 0.8636*** (-0.0195) |
| w/o akNN & Hypergraph CL | 80.24%*** (-6.23%) | 0.6836*** (-0.0613) | 0.9696*** (-0.0126) | 0.7980*** (-0.0418) | 91.83%*** (-0.78%) | 0.7817*** (-0.0295) | 0.9826*** (-0.0082) | 0.8620*** (-0.0211) |
| Baseline GCN | 79.53%*** (-6.94%) | 0.6739*** (-0.0710) | 0.9671*** (-0.0151) | 0.7912*** (-0.0486) | 91.74%*** (-0.87%) | 0.7765*** (-0.0347) | 0.9810*** (-0.0098) | 0.8583*** (-0.0248) |

The contents in parentheses denote the performance difference compared to the Full MMIHCL setup. Statistical significance of the differences between each setup and Full MMIHCL was determined by two-sided Wilcoxon signed-rank tests (***: $P < 0.001$, **: $P < 0.01$, *: $P < 0.05$, ns: not significant).

GCN-aggregated embeddings for intra-modality feature aggregation while feeding unpurified cross-modality matching pairs into the joint learning phase.

7. **w/o akNN & Hypergraph CL**: This setup replaces the adaptive neighbor selection with a constant $k$ and completely discards the hypergraph contrastive learning framework. It restricts the model to a rigid intra-modality graph topology and the sum of the original embeddings and GCN-aggregated embeddings for feature aggregation, while still feeding purified cross-modality matching pairs into the joint learning phase.

8. **Baseline GCN (Triple-module Removal)**: This represents a conventional graph learning baseline — effectively serving as the complete triple-module removal scenario — that operates without the reliability filter and utilizes a constant $k$ for graph construction. It directly feeds the raw embeddings into a standard GCN to obtain the aggregated representations for subsequent steps, serving as a reference for the synergistic effectiveness of our proposed dual-view architecture.

Table 1 presents the quantitative comparison (see S1 Table for detailed results across all 8 datasets and 10 metrics). In the weak linkage scenario, Full MMIHCL achieves the best comprehensive performance ($S_{overall}$) compared to the Baseline GCN and individual module removal variants. Crucially, it yields the highest ACC, demonstrating that the learned embeddings effectively preserve cell-type distinguishability for accurate alignment. Notably, the "w/o Hypergraph with CL" setup exhibits a significant decline in $S_{bio}$ ($P < 0.001$) despite a marginal increase in $S_{batch}$. This trade-off suggests that relying solely on the simple graph view leads to over-smoothing, where distinct cell populations are merged to achieve artificial mixing at the expense of biological distinctness. In the strong linkage scenario, while performance gaps narrow due to better data quality, this pattern persists ($P < 0.01$ for $S_{bio}$ drop), confirming that our dual-view architecture is essential for balancing batch correction with the preservation of biological heterogeneity.

Furthermore, to explicitly investigate the synergistic effects among the core components, we analyzed the performance variations under pairwise and complete triple-module removals. The results demonstrate a profound complementary synergy rather than simple additive effects. For instance, in the weak linkage scenario, while the combined removal of the filter with the akNN or Hypergraph CL module leads to severe degradations ($\Delta S_{overall}$ = –0.0335 and –0.0390, respectively), the most significant performance drop occurs when both the akNN and Hypergraph CL modules are removed simultaneously ($\Delta S_{overall}$ = –0.0418, with a 6.23% decrease in ACC). This indicates that the high-order structural information captured by Hypergraph CL fundamentally relies on the accurate intra-modality local geometries constructed by the akNN module. This mutual reinforcement contributes fundamentally to the model's resilience, yielding a dominant specific contribution ratio of 86.01% (compared to 68.93% and 80.25% for other pairwise combinations) to the overall performance improvement over the baseline ($\Delta S_{overall}$ = –0.0486 for Baseline GCN), which prevents integration collapse under extreme inter-modality noise. Consequently, this highly coupled, closed-loop system — rather than the mere stacking of individual components — serves as the core innovation that empowers MMIHCL to robustly handle challenging datasets.

Having established the quantitative superiority of the full MMIHCL model, we investigate the akNN mechanism using the CITE-seq PBMC dataset [28] as a representative example (consistent trends for the remaining seven datasets are presented in S8 Fig). To strictly isolate the akNN module's contribution, we applied it directly to the preprocessed count matrices of the primary modality (specifically RNA for the CITE-seq PBMC dataset, while utilizing RNA or high-dimensional protein for others) for each dataset, extracting the learned neighbor counts ($k$) based on ground truth. We deliberately excluded the final joint embeddings to prevent the complex iterative effects of hypergraph learning and CCA from obscuring the results. This ensures that the related visual results can purely reflect the module's adaptive response to intrinsic data heterogeneity.

Fig 8a reveals severe class imbalance, ranging from Monocytes 30.33% to Dendritic Cells 2.34%, where a fixed $k$ would inevitably cause over-smoothing or under-denoising. To counter this, akNN dynamically adjusts the receptive field. Fig 8b confirms a strong positive Spearman's Rank correlation ($R_S$ = 0.9386, $P < 0.001$) between cluster size and learned $k$ values (derived from an initial $k = 30$): smaller neighborhoods are assigned to rare types to preserve local topology, while larger $k$ values enhance connectivity in abundant populations. This adaptive strategy is further elucidated in Fig 8c, which employs Kernel Density Estimation (KDE) to visualize the distribution of $k$ values. By independently normalizing the density curves for each cell type, the plot highlights distinct, non-overlapping modes, confirming that the model learns specific structural requirements for different populations. Finally, Fig 8d serves as visual validation. The annotated UMAP demonstrates that despite these varying $k$ assignments, the model successfully maintains clear boundaries for rare clusters (like DCs) without merging them, proving that akNN effectively balances local preservation with global integration.

## Analysis on sensitivity and robustness

To comprehensively evaluate the stability of MMIHCL, we conducted a systematic sensitivity analysis on three key hyperparameters ($T, k, \lambda$) and a robustness test on the linkage ratio ($\rho$). All detailed experimental results are visualized in S9 Fig.

We firstly utilized the CITE-seq PBMC [28] and CyTOF human H1N1 & IFNG [34,35] datasets as representative benchmarks for weak and strong linkage scenarios, respectively. Considering the iterative optimization rounds ($T$), it is important to note that the performance at $T = 0$ serves as a baseline derived solely from the initial matching and CCA-based embeddings, as the iterative optimization process has not yet commenced. The results indicate a substantial performance leap from $T = 0–1$, confirming the necessity of our iterative strategy. The gains saturate around $T = 3$, where the model achieves peak $S_{bio}$ scores of 0.8483 and 0.8589 for two datasets. This rapid convergence can be largely attributed to the robust initialization strategy (detailed in Materials and methods, MMIHCL pipeline, Stage 2), which provides a high-quality starting point for the subsequent optimization. Since extending optimization beyond this point ($T > 3$) yields negligible improvements or even slight overfitting, we set $T = 3$ as the default to balance efficiency and performance.

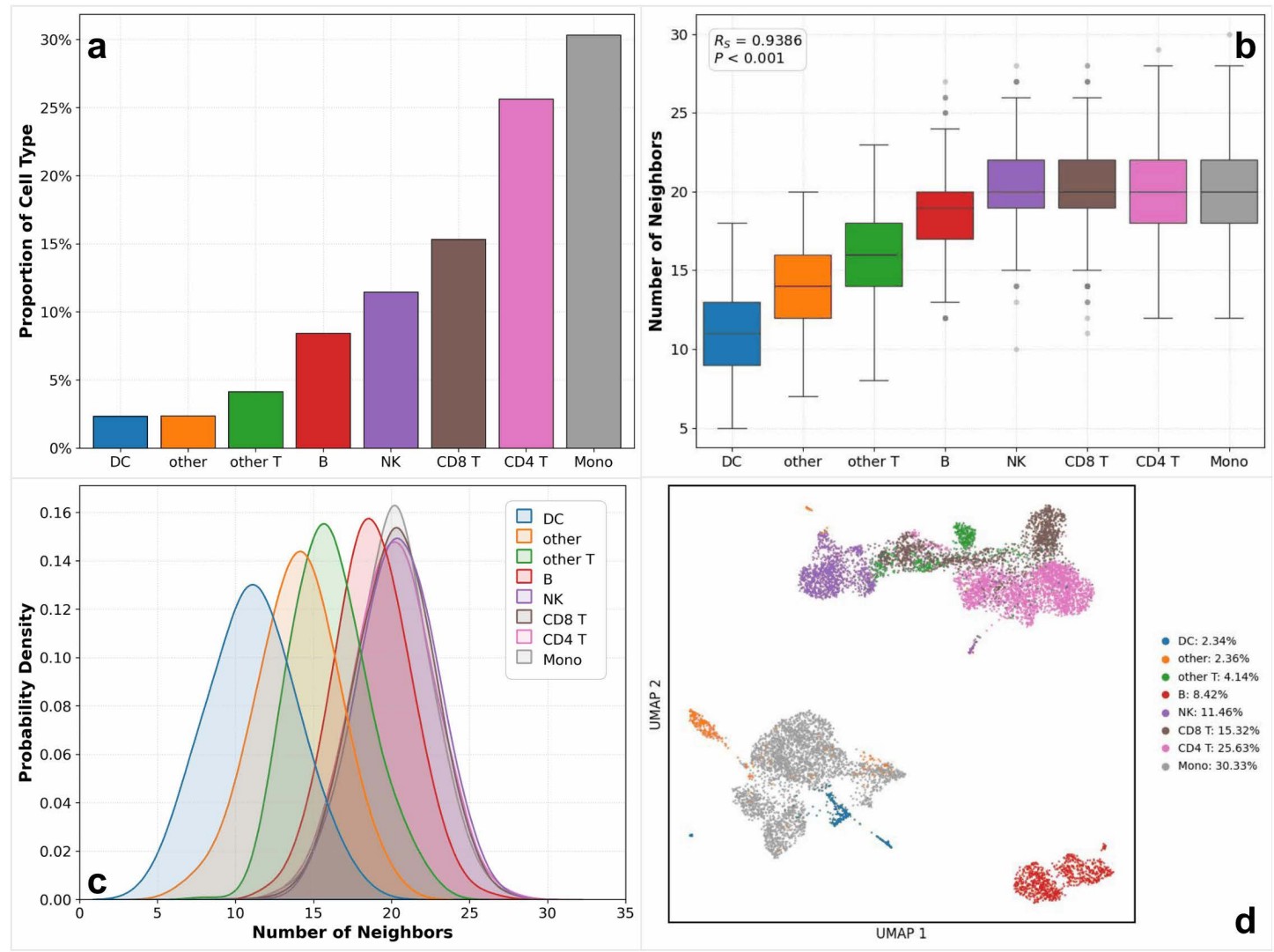

**Fig 8. Comprehensive analysis of the akNN mechanism. (a)** Distribution of cell type proportions in the CITE-seq PBMC dataset. **(b)** Box plots of the learned neighbor counts ($k$) across cell types, with Spearman correlation ($R_s$) indicating the relationship between cluster size and $k$. **(c)** Density estimation of learned $k$ values for each cell type using KDE. Curves are independently normalized to visualize distribution shapes across varying population sizes. **(d)** UMAP visualization labeled with the specific proportion of each cell population.

Regarding the other hyperparameters, we observed that $k$ and $\lambda$ exhibit broad stability ranges, though their roles are distinct. The neighbor count $k$ determines the granularity of the local graph structure. While an extremely small $k$ might fail to capture sufficient local topology, our results show that integration performance remains robust within $k \in [15, 25]$. Similarly, $\lambda$ controls the smoothness of the transformation. Although very large values can over-constrain the model, MMIHCL maintains consistent $S_{overall}$ scores within $\lambda \in [0.001, 0.1]$, indicating that the method effectively prevents overfitting without requiring exhaustive fine-tuning.

We further assessed the robustness of MMIHCL to linkage sparsity. For this analysis, we selected the CyTOF human H1N1 & IFNG and 10X-Multiome PBMC [36] datasets. These two datasets were chosen because they represent distinct strong linkage modality combinations (protein & protein and RNA & ATAC), allowing us to reliably simulate "weak linkage"

scenarios by artificially downsampling their linked features. We varied the linkage ratio $\rho$ from 100% down to 10% for both datasets, and further extended the stress test to 1% for the second dataset given its high feature dimensionality. Despite scarce shared information, MMIHCL avoided catastrophic collapse. On the 10X-Multiome PBMC dataset with only 1% linked features, $S_{overall}$ remained at 0.6547, retaining over 73.30% of the full-linkage performance. Crucially, $S_{batch}$ remained high at 0.8996 even at $\rho = 1\%$, confirming that MMIHCL can effectively leverage extremely limited shared information to guide alignment. Simultaneously, on the CyTOF human H1N1 & IFNG dataset at $\rho = 10\%$, MMIHCL retained over 89.86% of its baseline performance.

## Analysis on computational cost

In addition to integration performance, we compared the time and space efficiency of MMIHCL with other baselines. In this study, all experiments were run in one server with 512 GB RAM memory, 24 cores of Intel(R) Xeon(R) Gold 6248R CPU 3.00GHz, and a NVIDIA GeForce RTX 4090 GPU with 24 GB memory.

We firstly benchmarked the Running Time (RT) and Memory Consumption (MC) on eight datasets used in the previous performance assessment (as shown in Fig 9a and 9b). MMIHCL demonstrated high efficiency, matching fast methods like CelLink and MaxFuse, while significantly outperforming computationally intensive baselines such as MARIO ($\Delta = +28.8687$ minutes, $P<0.001$) and scConfluence ($\Delta = +12.5800$ minutes, $P<0.001$). Critically, regarding memory consumption, MMIHCL achieved exceptionally low peak usage (1.0866 GB on average) among all methods, surpassing even the non-deep learning baseline Seurat ($\Delta = +0.3233$ GB, $P<0.001$). Notably, the recently proposed scMRDR exhibited an unparalleled spatio-temporal efficiency across these benchmarks, achieving an even lower average running time (6.8312 minutes) and an extremely minimal memory usage (0.0475 GB).

We further investigated scalability by varying dataset sizes from 10k to 100k cells (as shown in Fig 9c and 9d), generated via bootstrap sampling from the CITE-seq PBMC dataset [28]. To ensure robust measurement, we recorded the average values of five independent runs for datasets $\leq$ 30k cells and a single run for larger scales, imposing strict cutoffs at 4 hours for RT and 64 GB for MC. Results showed that while MARIO and scConfluence timed out at 70k and 100k cells respectively, MMIHCL exhibited stable linear scalability, completing the 100k-cell integration in 208.84 minutes. In terms of space complexity, unlike CelLink triggered "out of Memory" errors (>64 GB) at 100k cells, MMIHCL remained extremely lightweight. At 100k cells, MMIHCL consumed only 2.16 GB, which is not only comparable to the highly efficient MaxFuse (1.80 GB), but also significantly lower than Seurat (8.04 GB) and other deep learning baselines. Furthermore, we must acknowledge the powerful scalability of scMRDR: it ranked third in processing speed on the 100k dataset (completed in 77.7147 minutes) and maintained a near-zero memory overhead (0.3524 GB) across all data scales, demonstrating exceptional efficiency for large-scale integration project.

## Discussion

By integrating multimodal information of the same cell, single-cell multimodal integration method can analyze cell state and function more comprehensively. Most existing methods focus on the scenarios of strong linkage dataset integration, but their performance on weak linkage datasets degrade to different degrees, so they cannot meet the growing need of high-performance weak linkage dataset integration. In this study, based on the current challenge of weakly linked single-cell multimodal integration, we propose MMIHCL, an unsupervised deep learning method through hypergraph contrastive learning designed to not only address integration bottlenecks but also empower downstream biological analysis.

MMIHCL's pipeline is built on several advances to overcome the difficulty of integrating weak linkage datasets. First, for high-quality local message passing, MMIHCL uses adaptive kNN to flexibly adjust the cell neighborhoods, and uses hypergraph embedding learning to integrate higher-order relationship between cells from single-cell data. Second, by taking advantage of linked and all multimodal features, and iteratively optimizing single-cell matching pairs and joint embeddings, MMIHCL can explore and fuse information from multimodal data adequately. Message propagation within

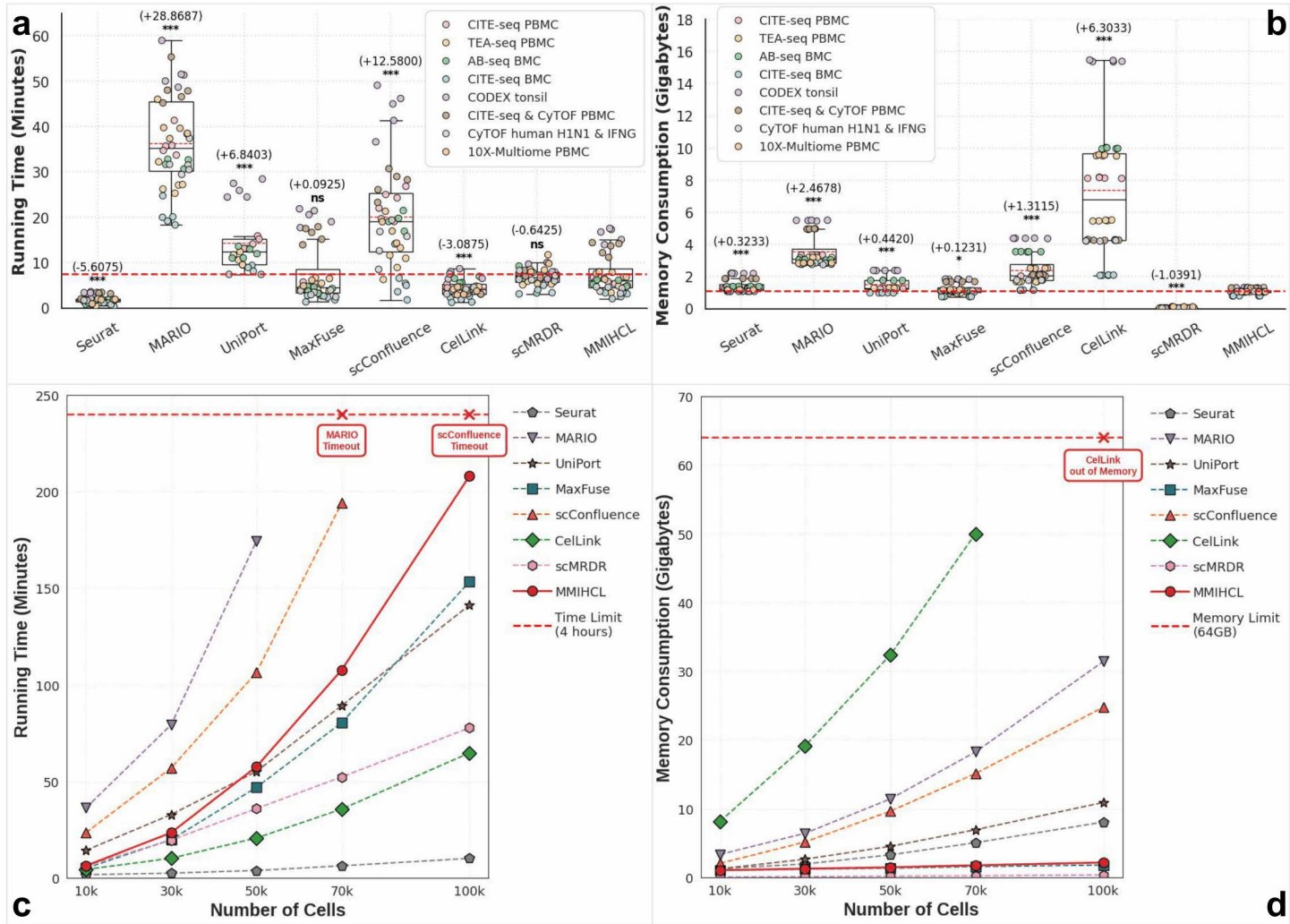

**Fig 9. Comprehensive analysis of computational cost and scalability. (a)** Benchmarking of Running Time (RT) in minutes across datasets. **(b)** Benchmarking of peak Memory Consumption (MC) in Gigabytes (GB). **(c, d)** Scalability analysis of RT and MC on datasets ranging from 10k to 100k cells, while the red dashed line in (c) and (d) indicates the 4-hour time and 64GB memory limit respectively.

and between modalities ensures that the joint embeddings generated by MMIHCL capture as much valuable information as possible.

For weak linkage scenarios, MMIHCL consistently delivers high-quality integration, maintaining a statistically significant lead over SOTA methods in weighted overall scores across diverse transcriptomic and proteomic datasets. For strong linkage scenarios, MMIHCL retains the highest performance ceiling, optimally balancing biological fidelity and multimodal alignment. For cross-modality feature prediction, MMIHCL achieves high-fidelity reconstruction of missing modalities, leveraging explicit cell matching to effectively bridge the information gap between disparate omics layers. For downstream biological discovery, MMIHCL empowers robust disease classification and drug target discovery, utilizing optimized joint embeddings to recover subtle pathological signals from background noise.

Beyond standard integration benchmarks, the practical applicability of a method depends heavily on its structural robustness and computational scalability. Our comprehensive evaluation across diverse tasks confirms MMIHCL as a

versatile solution for multimodal integration. The ablation study rigorously validated the critical roles of the adaptive neighbor selection and dual-view hypergraph learning in preventing over-smoothing and handling class imbalance. Additionally, the sensitivity analysis confirmed MMIHCL's stability across hyperparameter variations and resilience to sparse linkage, ensuring reliable integration without exhaustive fine-tuning. Furthermore, the computational analysis demonstrated MMIHCL's superior efficiency, highlighting its linear scalability and minimal memory footprint compared to SOTA baselines. Taken together, these analyses demonstrate that MMIHCL is not only accurate but also structurally reliable and computationally efficient for large-scale applications.

Although MMIHCL presents distinct advantages over existing methods in both technical alignment and downstream discovery, it still has some limitations to be addressed in future work. First, regarding the integration of datasets with significant biological differences, MMIHCL currently relies on the intersection of common cell types and lacks a dedicated mechanism to explicitly detect novel cell populations in the query data. Second, the current framework operates in a transductive manner and does not yet support transfer learning. This means that extending MMIHCL to online integration scenarios—where new samples are mapped onto a pre-trained reference atlas without retraining—remains a challenge. In the future, incorporating inductive inference capabilities and novelty detection modules will further elevate MMIHCL from a robust integration tool to a scalable engine for building and updating comprehensive single-cell multi-omics atlases.

## Materials and methods

### MMIHCL pipeline

**Stage 1: input and graph construction.** Suppose that two preprocessed datasets are denoted as $\mathbf{X} \in \mathbb{R}^{n_x \times p_x}$ and $\mathbf{Y} \in \mathbb{R}^{n_y \times p_y}$, where $\mathbf{X}$ consists of $n_x$ cells and $p_x$ features, $\mathbf{Y}$ consists of $n_y$ cells and $p_y$ features. MMIHCL finds a matching cell in $\mathbf{Y}$ for each cell in $\mathbf{X}$. There are $p_{link} \leq \min\{p_x, p_y\}$ linked features between $\mathbf{X}$ and $\mathbf{Y}$, which can be obtained from the prior knowledge between the two modalities.

To quantitatively measure the linkage intensity between modalities, we define the linkage ratio $\rho$ for each dataset as:

$$\rho = \min(\frac{p_{link}}{p_x}, \frac{p_{link}}{p_y}).$$

(1)

Based on these ratios, we categorize the integration scenarios into weak and strong linkage. A scenario is defined as weak linkage if the linked features represent a small fraction of at least one modality, i.e., $\rho < 10\%$. Conversely, a scenario is defined as strong linkage if both modalities share a substantial portion of linked features, i.e., $\rho \geq 10\%$.

For how to acquire these linked features, we use the prior interaction knowledge between ATAC regions and genes, genes and proteins, and same type of modal data to get linked features. If an ATAC region located near the promoter of a gene, that is, this region lies within the 2000 base-pair upstream activation region or the gene body, then we link these two features. For all proteins, we link each of them with their corresponding encoding genes. For same modal data, we take the intersection of their feature sets, and link one-to-one features with the same name.

Thus, $\mathbf{X}$ and $\mathbf{Y}$ have $p_{x,dist} = p_x - p_{link}$ and $p_{y,dist} = p_y - p_{link}$ modal-specific features respectively. We can rewrite these two datasets as horizontal concatenations of the linked and modal-specific feature parts as below:

$$\mathbf{X} = (\mathbf{X_{link}}, \mathbf{X_{dist}}) \in \mathbb{R}^{n_x \times (p_{link} + p_{x,dist})},$$
$$\mathbf{Y} = (\mathbf{Y_{link}}, \mathbf{Y_{dist}}) \in \mathbb{R}^{n_y \times (p_{link} + p_{y,dist})}.$$

Considering that similar cells in the feature space usually correspond to similar cell types, we convert the feature space information into an weighted adjacent cell graph. Let's take $\mathbf{X}$ as an example to illustrate the construction process of its adjacent cell graph $\mathbf{A_x}$.

Considering computational efficiency, we first use the PCA [39] dimensionality reduction method to preserve $p'_x$ principal components of $\mathbf{X}$ to get $\mathbf{X}' \in \mathbb{R}^{n_x \times p'_x}$. We then use $\mathbf{X}'$ to construct the adjacent cell graph $Graph_x = (Ver_x, Edge_x)$ , where $Ver_x$ represents the set of $n_x$ cells, and the embeddings of $Ver_x$ are $n_x$ embeddings of $\mathbf{X}'$. In addition, $Edge_x$ of $Graph_x$ represents the set of connected weighted edges between cells. Here the weight $W \in [0, 1]$ of the edge represents the correlation of feature vectors between cells, and greater weight means higher correlation.

First we apply the kNN [40] algorithm on $\mathbf{X}'$, then choose the top-$k$ nearest cells based on edge weights as its neighbors for each cell to get edges $\{(e_{i1}, e_{i2}, w_i)\}$, where $(e_{i1}, e_{i2})$ represents the two vertices of edge $i$, and $w_i$ represents its weight. For each cell $i$, the correlation to its $k$ nearest neighbors are sorted in an descending order ($w_{i1} \geq w_{i2} \geq \cdots \geq w_{ik}$), we next need to dynamically adjust the $k$ value based on the $k$ correlation distribution.

For cell $i$, the cell $j$ should be a neighbor of $i$ with probability $s_{ij}$. We compute this probability by minimizing the following objective function:

$$\min \sum_{i=1}^{n_x} \sum_{j=1}^{n_x} \sum_{m=0}^{+\infty} (1 - w_{ij}) s_{ij}^m,$$

$$\text{subject to } \forall i, 0 \leq s_{ij} \leq 1, \mathbf{s}_i^T \mathbf{1}_{n_x} = 1. \tag{2}$$

In Eq (2), $\mathbf{1}_{n_x}$ represents the $n_x$-dimensional vector whose elements are all 1. By introducing the standard Karush-Kuhn-Tucker condition [41], we can get the optimal solution of Eq (2):

$$s_{ij} = \begin{cases} 1, & \text{if } w_{ij} > 1 - \left( \frac{\sum_{l=1}^{k} \sqrt{w_{il}}}{k-1-\delta} \right)^2; \\ 0, & \text{otherwise.} \end{cases} \tag{3}$$

where $\delta \leq 0$ is a hyperparameter to control the sensitivity to the local distance change. Therefore, the number of adjusted nearest neighbors of cell $i$, denoted as $k_i$, is:

$$k_i = \begin{cases} k, & \text{if } w_{ik} > 1 - \left( \frac{\sum_{l=1}^{k} \sqrt{w_{il}}}{k-1-\delta} \right)^2; \\ j, & \text{if } w_{ij} > 1 - \left( \frac{\sum_{l=1}^{k} \sqrt{w_{il}}}{k-1-\delta} \right)^2 \text{ and} \\ & w_{i,(j+1)} \leq 1 - \left( \frac{\sum_{l=1}^{k} \sqrt{w_{il}}}{k-1-\delta} \right)^2. \end{cases} \tag{4}$$

From that we can get the new $Edge_x$, where the number of total edges is determined by the $k$ correlation distribution of each cell. The details of this adjustment process can be found in [32,42,43].

Now we can get the weighted adjacent matrix $\mathbf{A_x} \in [0, 1]^{n_x \times n_x}$, where the non-zero element $[\mathbf{A_x}]_{ij}$ represents the correlation strength between cell $i$ and $j$. To facilitate the following Graph Convolutional Network (GCN) operation, the normalized adjacency matrix $\bar{\mathbf{A}}_\mathbf{x}$ is calculated as:

$$\bar{\mathbf{A}}_\mathbf{x} = \mathbf{D}_\mathbf{x}^{-1/2} (\mathbf{A_x} + \mathbf{I}_{n_x}) \mathbf{D}_\mathbf{x}^{1/2}. \tag{5}$$

$\mathbf{D_x}$ in Eq (5) represents a $n_x \times n_x$ degree matrix whose diagonal entries are the degrees of vertices, where $[\mathbf{D_x}]_{ii} = \Sigma_j [\mathbf{A_x} + \mathbf{I}_{n_x}]_{ij}$. $\mathbf{I}_{n_x}$ in Eq (5) represents identity matrix of order $n_x$. The construction of $\bar{\mathbf{A}}_\mathbf{y}$ (the normalized adjacent matrix of $\mathbf{Y}$) is the same as that of $\bar{\mathbf{A}}_\mathbf{x}$.

**Stage 2: hypergraph representation learning and initial matching.** Given an embedding matrix $\mathbf{E} \in \mathbb{R}^{n \times d}$ and its normalized adjacent matrix $\bar{\mathbf{A}} \in [0, 1]^{n \times n}$, we refer to the process of hypergraph embedding learning on $\mathbf{E}$ as

$\hat{\mathbf{E}} \leftarrow H(\mathbf{E}, \bar{\mathbf{A}}) \in \mathbb{R}^{n \times d}$. The function $H(\cdot)$ is a $L$-layer structure, where each layer has a local and global message passing module respectively, and two modules in the same layer are related by contrastive learning. We denote the $n$ embeddings of $\mathbf{E}$ as $\boldsymbol{e_1}, \boldsymbol{e_2}, \cdots, \boldsymbol{e_n} \in \mathbb{R}^d$, and use $\mathbf{E}$ as layer 0 embeddings of $H(\cdot)$, that is, $\mathbf{E}^{(0)}$.

We firstly introduce the local message passing module implemented through GCN. Given the layer 0 embedding matrix $\mathbf{E}^{(0)}$ and set the layer 0 local embedding branch $\bar{\mathbf{E}}^{(0)} = \mathbf{E}^{(0)}$, the message passing process in layer $l$ can be expressed as:

$$\bar{\mathbf{E}}^{(l)} = \bar{\mathbf{A}}\mathbf{E}^{(l-1)}. \tag{6}$$

We secondly introduce the global message passing module. Let $\mathbf{W} \in \mathbb{R}^{d \times K}$ be a dynamically learnable parameter matrix containing high order internal relationships between $n$ embeddings of $\mathbf{E}$. Given the layer 0 global embedding branch $\tilde{\mathbf{E}}^{(0)} = \mathbf{E}^{(0)}$, we formulate the message passing process on the learnable parameter of layer $l$ as:

$$\tilde{\mathbf{E}}^{(l)} = \mathbf{H}\mathbf{H}^T\mathbf{E}^{(l-1)} = (\mathbf{E}^{(0)}\mathbf{W})(\mathbf{E}^{(0)}\mathbf{W})^T\mathbf{E}^{(l-1)}. \tag{7}$$

This design utilizes $\mathbf{H} = \mathbf{E}^{(0)}\mathbf{W}$ as a "soft" incidence matrix mapping $n$ cells to $K$ virtual hyperedges. The operation $\mathbf{H}(\mathbf{H}^T\mathbf{E}^{(l-1)})$ executes the "node-hyperedge-node" paradigm: $\mathbf{H}^T$ aggregates features into latent hyperedge space, and $\mathbf{H}$ redistributes high-order context back to cells. Dynamically learning $\mathbf{w}$ captures global dependencies transcending initial local constraints, ensuring robustness for complex, weakly linked multimodal distributions.

Contrastive learning aligns both branches at each layer to ensure $\mathbf{H}$ captures biologically relevant dependencies. By maximizing agreement between the local (neighborhood-preserving) and global (high-order manifold) branches ($\boldsymbol{e}_i^{\overline{(l)}}$, $\boldsymbol{e}_j^{\widetilde{(l)}}$ for $i, j = 1, 2, \cdots, n$ and $l = 1, 2, \cdots .L$), $\mathcal{L}_c$ grounds the implicit hypergraph in cell identities while exploring global structures. The objective function is defined as:

$$\mathcal{L}_c = -\sum_{i=1}^{n}\sum_{l=1}^{L} \log \frac{\exp[\text{sim}(\boldsymbol{e}_i^{\overline{(l)}}, \boldsymbol{e}_i^{\widetilde{(l)}})/\tau]}{\sum_{j=1}^{n} \exp[\text{sim}(\boldsymbol{e}_i^{\overline{(l)}}, \boldsymbol{e}_j^{\widetilde{(l)}})/\tau]}. \tag{8}$$

In Eq (8), $\text{sim}(\cdot)$ represents cosine similarity calculation function, and $\tau$ denotes the temperature hyperparameter to control the penalties on hard negative samples.

Then, we combine the local and global branches to get the embeddings of layer $1 \sim L$, and sum them to get the final output of $H(\cdot)$:

$$\mathbf{E}^{(l)} = \bar{\mathbf{E}}^{(l)} + \tilde{\mathbf{E}}^{(l)}, \quad \hat{\mathbf{E}} = \sum_{l=1}^{L} \mathbf{E}^{(l)}. \tag{9}$$

We adopt direct summation based on two considerations. First, the local and global representations are structurally complementary. Crucially, the contrastive learning part explicitly aligns these views in a shared latent space. By minimizing the contrastive loss ($\mathcal{L}_c$), the model is driven to harmonize the feature distributions, which naturally balances the magnitudes of the local and global embeddings. Second, adhering to the principle of parsimony (Occam's razor), we employ unweighted summation as a robust, parameter-free mechanism. This avoids introducing additional hyperparameters for weight tuning, thereby reducing model complexity and mitigating overfitting risks on sparse single-cell data.

To prevent the model from overfitting, we introduce the regularization term into the final objective loss function as follows:

$$\mathcal{L} = \mathcal{L}_c + \lambda\|\Theta\|_2^2, \tag{10}$$

where $\lambda$ is the weight decay coefficient for regularization term, and $\Theta$ is the parameters of the model.

After obtaining the linked feature datasets $(\mathbf{X_{link}}, \mathbf{Y_{link}})$ and normalized adjacency graphs $(\bar{\mathbf{A}}_\mathbf{x}, \bar{\mathbf{A}}_\mathbf{y})$ in stage 1, we can get two hypergraph embeddings:

$$\hat{\mathbf{X}}_{\mathbf{link}} \leftarrow H(\mathbf{X_{link}}, \bar{\mathbf{A}}_\mathbf{x}) \in \mathbb{R}^{n_x \times p_{link}},$$

$$\hat{\mathbf{Y}}_{\mathbf{link}} \leftarrow H(\mathbf{Y_{link}}, \bar{\mathbf{A}}_\mathbf{y}) \in \mathbb{R}^{n_y \times p_{link}}.$$

Then calculate the 1 - PCC between the row elements of $\hat{\mathbf{X}}_{\mathbf{link}}$ and $\hat{\mathbf{Y}}_{\mathbf{link}}$ to get the link feature correlation distance matrix $\mathbf{D}^{(0)} \in [0, 1]^{n_x \times n_y}$.

We obtain an initial matching group $\mathbf{\Pi}^{(0)} = (\mathbf{\Pi}_1^{(0)}, \mathbf{\Pi}_2^{(0)}, \cdots, \mathbf{\Pi}_p^{(0)})$, by solving the first $p$ optimal solutions of the following linear assignment problem [44]:

$$\text{minimize } \langle \mathbf{\Pi}, \mathbf{D} \rangle,$$

$$\text{subject to } \mathbf{\Pi} \in \{0, 1\}^{n_x \times n_y}, \ \mathbf{\Pi}\mathbf{1}_{n_y} = \mathbf{1}_{n_x}. \tag{11}$$

In Eq (11), $\langle \mathbf{\Pi}, \mathbf{D} \rangle = \sum_{i=n_x}^{n_y} \sum_{j=n_x}^{n_y} [\mathbf{\Pi}]_{ij} [\mathbf{D}]_{ij}$ denotes the trace inner product.

Given $\mathbf{D}^{(0)}$, under the constraint conditions of Eq (11), the initial matching group $\mathbf{\Pi}^{(0)}$ satisfies:

$$\mathbf{\Pi}_1^{(0)} = \text{argmin}_{\mathbf{\Pi}} \langle \mathbf{\Pi}, \mathbf{D}^{(0)} \rangle,$$

$$\mathbf{\Pi}_2^{(0)} = \text{argmin}_{\mathbf{\Pi} \backslash \{\mathbf{\Pi}_1^{(0)}\}} \langle \mathbf{\Pi}, \mathbf{D}^{(0)} \rangle,$$

$$\cdots$$

$$\mathbf{\Pi}_p^{(0)} = \text{argmin}_{\mathbf{\Pi} \backslash \{\mathbf{\Pi}_1^{(0)}, \mathbf{\Pi}_2^{(0)}, \cdots, \mathbf{\Pi}_{p-1}^{(0)}\}} \langle \mathbf{\Pi}, \mathbf{D}^{(0)} \rangle. \tag{12}$$

From that we can get $pn_x$ matching pairs $\mathbf{M}^{(0)} = \{(i, j, d_{ij})\}$, where $(i,j)$ represents matched cell index, and $d_{ij} \in [0, 1]$ represents matching degree (1 - PCC of embeddings between cells $i$ and $j$, smaller is better match). Considering that low quality matching may reduce the quality of subsequent integration, we need to set up a filter to provide quality control over the matching pairs $\mathbf{M}^{(0)}$.

In the filter, we only keep pairs such that $d_{ij} \in [0, \alpha]$, where $\alpha \in (0, 1]$ is a matching degree threshold. It is also necessary to ensure that every cell in $\mathbf{X}$ has at least one corresponding matching cell in $\mathbf{Y}$, so we have $n_x \leq |\mathbf{M}^{(0)}| \leq pn_x$. For simplicity, we will refer to the process of obtaining filtered $\mathbf{M}^{(0)}$ as a function $\mathbf{M}^{(0)} \leftarrow M(\hat{\mathbf{X}}_{\mathbf{link}}, \hat{\mathbf{Y}}_{\mathbf{link}})$.

**Stage 3: iteratively optimizing cell matching.** Given filtered initial matching pairs $\mathbf{M}^{(0)}$, we next transform $(\mathbf{X}, \mathbf{Y})$ into $(\mathbf{X^r}, \mathbf{Y^r})$ that fully integrates $(\mathbf{X}, \mathbf{Y})$ while preserving important original information about $(\mathbf{X}, \mathbf{Y})$. We use CCA to implement this transform process, which can be denoted as a function $(\mathbf{X^r}, \mathbf{Y^r}) \leftarrow C(\mathbf{X}, \mathbf{Y}, \mathbf{M}^{(0)})$.

We first choose the numbers of principal components $(r_x, r_y)$ to retain based on $(\mathbf{X}, \mathbf{Y})$, to obtain dimensionality reduction for these datasets, that is, $\mathbf{X^{r_x}} \in \mathbb{R}^{n_x \times r_x}$ and $\mathbf{Y^{r_y}} \in \mathbb{R}^{n_y \times r_y}$. Then we perform CCA on data pairs:

$$\{([\mathbf{X^{r_x}}]_i, [\mathbf{Y^{r_y}}]_j)\}, \text{for } i = 1, 2, \cdots, n_x; j = 1, 2, \cdots, n_y.$$

So we can obtain the leading $r$ loading vectors for these modalities, and collect them as columns for $\mathbf{C_x} \in \mathbb{R}^{r_x \times r}$ and $\mathbf{C_y} \in \mathbb{R}^{r_y \times r}$. Now we can get the initial joint embeddings $\mathbf{X^r} = \mathbf{X^{r_x}}\mathbf{C_x} \in \mathbb{R}^{n_x \times r}$ and $\mathbf{Y^r} = \mathbf{Y^{r_y}}\mathbf{C_y} \in \mathbb{R}^{n_y \times r}$.

Given initial matching pairs $\mathbf{M}^{(0)}$ obtained from stage 2 and all feature datasets $(\mathbf{X}, \mathbf{Y})$, we iteratively optimize the joint embeddings of these datasets. After setting the hyperparameters $\tau$, $\lambda$, etc., the entire iteration process is as follows for $t = 1, 2, \cdots, T$:

1. Compute the joint embeddings according to matching pairs from the last round: $(\mathbf{X}^{\mathbf{r},(\mathbf{t})}, \mathbf{Y}^{\mathbf{r},(\mathbf{t})}) \leftarrow C(\mathbf{X}, \mathbf{Y}, \mathbf{M}^{(\mathbf{t-1})})$.

2. Apply hypergraph embedding learning on this round of joint embeddings: $\hat{\mathbf{X}}^{\mathbf{r},(\mathbf{t})} \leftarrow H(\mathbf{X}^{\mathbf{r},(\mathbf{t})}, \bar{\mathbf{A}}_{\mathbf{x}})$, $\hat{\mathbf{Y}}^{\mathbf{r},(\mathbf{t})} \leftarrow H(\mathbf{Y}^{\mathbf{r},(\mathbf{t})}, \bar{\mathbf{A}}_{\mathbf{y}})$. If currently $t = T$, terminate the iteration.

3. Use the hypergraph learning results to generate refined matching pairs: $\mathbf{M}^{(\mathbf{t})} \leftarrow M(\hat{\mathbf{X}}^{\mathbf{r},(\mathbf{t})}, \hat{\mathbf{Y}}^{\mathbf{r},(\mathbf{t})})$.

In the round $T$ of iterative optimization, we only get the hypergraph learned embeddings $\hat{\mathbf{X}}^{\mathbf{r},(\mathbf{t})} \in \mathbb{R}^{n_x \times r}$, $\hat{\mathbf{Y}}^{\mathbf{r},(\mathbf{t})} \in \mathbb{R}^{n_y \times r}$ and the iteration is terminated. Now we use $(\hat{\mathbf{X}}^{\mathbf{r},(\mathbf{t})}, \hat{\mathbf{Y}}^{\mathbf{r},(\mathbf{t})})$ to generate the final matching $\mathbf{\Pi}^*$.

Suppose the user wants to find a one-to-N ($N = 1, 2, \cdots$ and $Nn_x \leq n_y$) matching between $\mathbf{X}$ and $\mathbf{Y}$, the final matching is no longer a group, but only the optimal solution of Eq (11). When $N = 1$, we solve Eq (11) directly to yield $\mathbf{\Pi}^* = \mathrm{argmin}_{\mathbf{\Pi}} \langle \mathbf{\Pi}, \mathbf{D}^{(\mathbf{T})} \rangle$, where $\mathbf{D}^{(\mathbf{T})}$ is the 1 - PCC distance matrix of $\hat{\mathbf{X}}^{\mathbf{r},(\mathbf{t})}$ and $\hat{\mathbf{Y}}^{\mathbf{r},(\mathbf{t})}$. When $N > 1$, we make $N$ copies of $\mathbf{D}^{(\mathbf{T})}$, and then concatenate them vertically to get $\mathbf{D}'^{(\mathbf{T})}$, next yield $\mathbf{\Pi}^* = \mathrm{argmin}_{\mathbf{\Pi}} \langle \mathbf{\Pi}, \mathbf{D}'^{(\mathbf{T})} \rangle \in [0, 1]^{Nn_x \times n_y}$. In this way, every cell in $\mathbf{X}$ corresponds to $N$ matches in $\mathbf{Y}$.

We extract the matching pairs in $\mathbf{\Pi}^*$ to get $\mathbf{M}^*$. After setting the joint embedding in dimension $r^*$, the final joint embeddings are $(\mathbf{X}^*, \mathbf{Y}^*) \leftarrow C(\mathbf{X}, \mathbf{Y}, \mathbf{M}^*)$ where $\mathbf{X}^* \in \mathbb{R}^{n_x \times r^*}$ and $\mathbf{Y}^* \in \mathbb{R}^{n_y \times r^*}$.

**Pseudo-code of MMIHCL pipeline.** For ease of understanding, we show the pseudo-code of the MMIHCL pipeline in Algorithm 1.

### Algorithm 1 MMIHCL pipeline

```
Require: Input two modal datasets (X, Y).
Ensure: Final cell matching Π* and joint embeddings (X*, Y*).
1: Use akNN to generate cell graphs of X and Y to get adjacent matrices (Aₓ, Ay), and their normal-
   ized versions (Āₓ, Āy).
2: Learn hypergraph-based embeddings from linked features, X̂_link ← H(X_link, Āₓ) and Ŷ_link ← H(Y_link, Āy).
3: Solve the linear assignment problem to obtain the initial matching pairs M⁽⁰⁾ ← M(X̂_link, Ŷ_link).
4: Iterate over the following steps for t = 1, 2, · · ·, T:
   • Use CCA to get joint embeddings (Xʳ,⁽ᵗ⁾, Yʳ,⁽ᵗ⁾) ← C(X, Y, M⁽ᵗ⁻¹⁾).
   • Learn hypergraph-based embeddings from CCA results X̂ʳ,⁽ᵗ⁾ ← H(Xʳ,⁽ᵗ⁾, Āₓ) and Ŷʳ,⁽ᵗ⁾ ← H(Yʳ,⁽ᵗ⁾, Āy). If
t = T, break.
   • Obtain the refined matching pairs M⁽ᵗ⁾ ← M(X̂ʳ,⁽ᵗ⁾, Ŷʳ,⁽ᵗ⁾).
5: Get the final matching pairs from the hypergraph-based embeddings M* ← M(X̂ʳ,⁽ᵗ⁾, Ŷʳ,⁽ᵗ⁾).
6: Acquire the final joint embeddings through CCA method (X*, Y*) ← C(X, Y, M*)
```

### Experimental datasets

We conducted experiments on eight datasets to comprehensively investigate the integration performance of MMIHCL. These datasets cover various biological systems, such as human PBMCs and BMCs, and represent diverse integration challenges categorized into two primary scenarios:

1. **Weakly linked scenarios**: This scenario includes five datasets (such as CITE-seq PBMC [28] and TEA-seq PBMC [29]) characterized by comparatively weak feature correlation across disparate modalities.

2. **Strongly linked scenarios**: This includes three datasets (such as CITE-seq & CyTOF PBMC [28,33] and CyTOF human H1N1 & IFNG [34,35]) involving cross-platform or cross-modality alignments.

Beyond these benchmarking datasets, we incorporated two application-specific datasets, including HPAP [37] and Kang18 PBMC [38], characterized by distinct biological conditions to validate MMIHCL in downstream discovery tasks. Unlike the standard benchmarking datasets, these two datasets contain condition-specific batch information, serving as the basis for disease classification and drug target discovery experiments.

A statistical summary of all these datasets is provided in Table A in S1 File. For more detailed description of these datasets, preprocessing of different raw datasets, and programming hyperparameter settings, please refer to Sections 2–4 in S1 File.

### Evaluation metrics

To provide a comprehensive and multi-dimensional assessment of MMIHCL, we employed seven quantitative metrics encompassing three essential properties of multimodal integration: matching accuracy, biological conservation, and multimodal alignment. This expanded framework follows the scIB benchmarking standards [45], aligning with recent SOTA protocols for multimodal integration [46,47], to ensure a standardized evaluation, as described below:

1. **Matching Accuracy**: This dimension evaluates the correctness of cell-cell matching. We utilized ACCuracy (ACC) as the primary metric. Additionally, the FOSCTTM was employed to measure alignment precision within the joint embedding space.

2. **Biological Conservation**: To assess the preservation of underlying biological heterogeneity, we employed the Normalized Mutual Information (NMI), Adjusted Rand Index (ARI), and Average Silhouette Width for cell type labels ($ASW_{label}$). These metrics ensure that the integrated embeddings retain distinct biological signals.

3. **Multimodal Alignment**: We evaluated the effectiveness of modality mixing using Graph Connectivity (GC) and Average Silhouette Width for batches ($ASW_{batch}$). These metrics quantify the mixed effect of integrated embeddings at both the local manifold level and the global embedding level.

The metrics defined above are not evaluated in isolation. Rather, they are nested within a weighted synthesis logic to form the overall score ($S_{overall}$). This score aggregates the multi-dimensional performance into a single indicator, reflecting the trade-off between biological fidelity (including matching accuracy and conservation) and batch effect removal (corresponding to multimodal alignment). The computation logic is defined in Eq (13):

$$S_{overall} = 0.6 \times S_{bio} + 0.4 \times S_{batch},$$
$$S_{bio} = \frac{1}{N_{bio}} \left[ ACC + (1 - FOSCTTM) \times \mathbb{I}_{paired} + NMI + \max(0, ARI) + ASW_{label} \right],$$
$$S_{batch} = \frac{1}{2} \left( GC + ASW_{batch} \right).$$

(13)

where $\mathbb{I}_{paired}$ is an indicator function that equals 1 for paired datasets and 0 otherwise. As all constituent metrics are normalized to the range [0, 1], the resulting $S_{bio}$, $S_{batch}$, and $S_{overall}$ are strictly bounded within the interval [0, 1]. Detailed mathematical definitions for this nested framework are available in Section 5 in S1 File.

### Supporting information

**S1 File. Supporting information for "Single-cell data integration across weakly linked modalities".**
(PDF)

**S1 Table. Detailed quantitative results of the ablation study.** This file contains the mean values, standard deviations, and performance differences (compared to the full MMIHCL model) for all 10 evaluation metrics (including 7 basic metrics and 3 composite scores: $S_{bio}$, $S_{batch}$, and $S_{overall}$) across all 8 datasets (5 weak linkage and 3 strong linkage datasets).
(XLSX)

**S1 Fig. UMAP visualization on TEA-seq PBMC dataset.** The first and third row subgraphs are colored by data modality, and the second and fourth row subgraphs are colored by cell type. Other UMAP graphs in supporting information are also arranged in this way.
(PDF)

**S2 Fig. UMAP visualization on AB-seq BMC dataset.**
(PDF)

**S3 Fig. UMAP visualization on CITE-seq BMC dataset.**
(PDF)

**S4 Fig. UMAP visualization on CODEX tonsil dataset.**
(PDF)

**S5 Fig. UMAP visualization on CITE-seq & CyTOF PBMC dataset.**
(PDF)

**S6 Fig. UMAP visualization on 10X-Multiome PBMC dataset.**
(PDF)

**S7 Fig. Performance evaluation of cross-modality feature prediction on the 10X-Multiome PBMC dataset.** (a) Violin plots displaying the distribution of cell-wise Pearson Correlation Coefficients (PCCs) between ground truth and predicted gene expression profiles. The white dot represents the median PCC, and the thick bar indicates the inter-quartile range. The numbers above the violins indicate the difference in median PCC relative to MMIHCL. Statistical significance was determined using two-sided Wilcoxon signed-rank tests (***: $P < 0.001$, **: $P < 0.01$, *: $P < 0.05$, ns: not significant). (b) CDF curves illustrating the proportion of cells (y-axis) surpassing specific PCC thresholds (x-axis). The translucent shading surrounding each curve represents the standard deviation, and the values in parentheses within the legend denote the AUC for each method. (c) Side-by-side heatmaps comparing the z-scored expression of representative marker genes (e.g., *MS4A1*, *CST3*, *CD8A*) across annotated cell types for ground truth, MMIHCL prediction, and MARIO prediction. Rows represent gene markers, and columns represent individual cells sorted by cell type. (d) UMAP visualizations of ground truth versus predicted expression for two lineage-specific gene markers: *MS4A1* (B cells) and *CST3* (Monocytes).
(PDF)

**S8 Fig. Analysis of cell type proportions and learned neighbor counts across seven additional datasets.** To analyze the neighbor distribution, the adaptive graphs were constructed using the primary modality of each dataset: CyTOF for CITE-seq & CyTOF PBMC, H1N1 for CyTOF human H1N1 & IFNG, and RNA for the remaining datasets. For each dataset, the bar plot illustrates the proportion of different cell types, highlighting the prevalence of class imbalance. The corresponding box plots display the distribution of adaptive $k$ values (derived from an initial $k = 30$) assigned by the akNN module. The Spearman's Rank correlation ($R_S$) and $P$-value annotated in each panel indicate a consistent and significant positive relationship between cluster size and neighbor counts across all datasets.
(PDF)

**S9 Fig. Analysis of hyperparameter sensitivity and robustness across three representative datasets.** Rows and columns correspond to four hyperparameters ($T$, $k$, $\lambda$, $p_{link}$) and three evaluation metrics ($S_{bio}$, $S_{batch}$, $S_{overall}$), respectively. Solid lines represent the mean of five independent runs, while shaded areas indicate the minimum-maximum range.
(PDF)

## Acknowledgments

We thank the National Supercomputer Center in Guangzhou (NSCG) at Sun Yat-Sen University (SYSU), for providing technical support.

## Author contributions

**Conceptualization:** Zhipeng Zhou.

**Data curation:** Zhipeng Zhou.

**Formal analysis:** Zhipeng Zhou.

**Funding acquisition:** Zhiming Dai.

**Investigation:** Zhipeng Zhou.

**Methodology:** Zhipeng Zhou.

**Project administration:** Zhiming Dai.

**Resources:** Zhipeng Zhou.

**Software:** Zhipeng Zhou.

**Supervision:** Yang Zhang, Zhiming Dai.

**Validation:** Zhipeng Zhou.

**Visualization:** Zhipeng Zhou.

**Writing – original draft:** Zhipeng Zhou.

**Writing – review & editing:** Zhipeng Zhou, Yang Zhang, Zhiming Dai.

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
