## [Decision Letter · Decision Letter 0]

18 Dec 2025

PCOMPBIOL-D-25-01825

Single-cell data integration across weakly linked modalities

PLOS Computational Biology

Dear Dr. Zhou,

Thank you for submitting your manuscript to PLOS Computational Biology. After careful consideration, we feel that it has merit but does not fully meet PLOS Computational Biology's publication criteria as it currently stands. Therefore, we invite you to submit a revised version of the manuscript that addresses the points raised during the review process.

We look forward to receiving your revised manuscript.

Kind regards,

Jifan Shi

Academic Editor

PLOS Computational Biology

Ilya Ioshikhes

Section Editor

PLOS Computational Biology

Additional Editor Comments :

The authors propose an interesting method, MMIHCL, for integrating weakly linked multimodal data. I have several comments:

1. The pipeline in Fig. 1 should be revised to improve readability, with less text and a clearer presentation of the workflow.

2. A rigorous and quantitative definition of “strong” versus “weak” linkage between modalities should be provided.

3. Several typographical issues remain (e.g., inconsistent use of quotation marks in the Author Summary), in addition to those already noted by other reviewers.

4. The statement that the method “outperforms state-of-the-art methods” is overused and insufficiently informative. Instead, the authors should emphasize the key conceptual contributions, empirical findings, and distinctive features of MMIHCL. While reporting performance gains (e.g., 1–4%) is necessary, such marginal improvements alone do not constitute a mechanistic or methodological innovation.

5. The authors suggest that MMIHCL can be applied to integrative studies such as disease classification and drug target discovery. To substantiate these claims, one or two real-world application examples should be included.

Journal Requirements:

3) Thank you for stating that "CyTOF PBMC: https://support.10Xgenomics.com/single-cell-gene-expression/datasets/3.0.2/5k_pbmc_protein_v3?". Please amend this to a working link.

4) Please amend your detailed Financial Disclosure statement. This is published with the article. It must therefore be completed in full sentences and contain the exact wording you wish to be published.

2) State what role the funders took in the study. If the funders had no role in your study, please state: "The funders had no role in study design, data collection and analysis, decision to publish, or preparation of the manuscript.".

Reviewers' comments:

Reviewer's Responses to Questions

Reviewer #1: The authors present MMIHCL, a deep learning framework designed to do diagonal integration for single-cell multimodal data. The authors benchmark MMIHCL against several state-of-the-art methods (Seurat, MARIO, UniPort, MaxFuse, scConfluence) across varying datasets (CITE-seq, TEA-seq, CODEX, etc.). They report superior performance in diagonal integration tasks, particularly in weakly linked scenarios.

The authors define "weak linkage" as scenarios where linked features comprise less than 10% of the total features. While this is a practical working definition, it would be beneficial to explore the lower bound of this performance. I recommend conducting a sensitivity analysis where the number of available linked features is artificially downsampled (e.g., using only 1%, 5%, 10% of shared genes/proteins) on a strong-linkage dataset, which would powerfully demonstrate the robustness of the "weak linkage" definition.

The authors propose an adaptive kNN (akNN) strategy (Stage 1) to address the challenge of unbalanced cell numbers across different cell types. The core premise is that the number of neighbors k should be dynamically adjusted for each cell based on local density, rather than using a fixed global k. While the ablation study (Table 1) indicates that removing akNN drops performance, there is no direct analysis showing how the adaptation behaves. To validate that the akNN is functioning as intended (i.e., adapting to cell density and cluster size), I recommend the authors provide a boxplot or histogram showing the distribution of the learned neighbor counts k for each cell grouped by cell type to better demonstrate the ability of this strategy.

The manuscript currently relies on four metrics (ACC, FOSCTTM, GC, and ARI) to validate the performance of MMIHCL. While these provide an initial view of matching accuracy and cluster purity, they do not fully capture the nuanced trade-off between biological conservation (preserving distinct cell states) and modality alignment (mixing the datasets). To align this study with current state-of-the-art standards in single-cell computational biology, I recommend expanding the evaluation to include a broader suite of metrics established in recent benchmarking studies. Specifically, I recommend the authors refer to recent published benchmarks on diagonal and mosaic integration (e.g., https://www.nature.com/articles/s41592-025-02856-3 and https://www.nature.com/articles/s41592-025-02737-9).

The manuscript currently evaluates the ability of the learned joint embeddings primarily through clustering. Since MMIHCL generates a unified, batch-corrected embedding space for inputs from different modalities, it would be compelling to test the predictive capacity of these embeddings. I recommend adding an experiment on a paired dataset (e.g., CITE-seq) to evaluate cross-modal feature prediction (such as training a regression to predict the masked omics layer using embedding).

The author should check and correct the typos across the manuscript, such as in L103-104, “MMIHCL achieves the highest ACCuracy (ACC) on all datasets with an average of 85.46%, surpassing MaxFuse and MMIHCL by 1.2% and 3.72%, respectively.” I think it should be “surpassing MaxFuse and” another method, not MMIHCL.

Reviewer #2: This paper addresses the core challenge of single-cell multimodal data integration in weakly correlated scenarios by proposing the deep learning framework MMIHCL based on hypergraph contrastive learning. Current single-cell sequencing technologies can obtain multimodal data such as transcriptomic and proteomic profiles, but the correlation features between emerging modalities and other modalities are scarce or weakly correlated, making it difficult for existing methods to accurately model intercellular relationships and learn effective cell representations. The core innovation of MMIHCL lies in integrating an optimized adaptive k-nearest neighbors (akNN) graph with a hypergraph contrastive learning framework through a three-stage process: first constructing modal-specific adjacency graphs, then generating initial cell matching pairs using correlation features, and finally optimizing the joint embedding and matching results through iterative CCA transformations and hypergraph learning. I have the following major comments:

1. The fusion method of local and global information transmission adopts direct summation, but the rationality of this method is not explained. There may be weight differences between local neighbor information and global high-order information, and simple summation may dilute key information.

2. During the iterative process, the subjectively defined algebraic T cannot guarantee optimal performance. Whether the desired outcome is achieved before the T-round or remains unfulfilled during the T-round.

3. The paper's visualizations and table layouts are poorly positioned. The weak linkage datasets and strong linkage datasets should be placed in more prominent locations for better readability.

4. In the Ablation studies section, the impact of individual modules was examined by modifying a single module to observe its effect. However, the combined effects of multiple modules might be difficult to determine. This challenge can be addressed by introducing multi-module modifications to assess their synergistic influence.

5. The study selected only two weak association integration methods—MaxFuse and scConfluence—as baselines, excluding other mainstream approaches published during the same period (including weak association integration tools introduced after 2023). This limitation may undermine the conclusion that MMIHCL achieves optimal performance' due to insufficient comparative evidence.

6. This study only evaluated computational costs under a single-server setup without comparing runtime and memory consumption across varying cell populations (e.g., 1k,10k,100k cells), thus failing to demonstrate the efficiency advantage of MMIHCL on large-scale datasets.

Reviewer #3: The authors present MMIHCL, a deep learning-based framework that leverages an optimized adaptive k-nearest neighbor graph to model single cell pair-wise relationships for multimodal data integration. MMIHCL uses hypergraph contrastive learning to capture the high-order information of a graph to produce cell representations. The authors conducted benchmark experiments on diverse strongly and weakly linked multimodal datasets to demonstrate the performance of MMIHCL and compare to other baseline methods in single-cell cross-modal integration and matching. The significance of the proposed method should be further demonstrated. Below are specific concerns.

Major:

1. Can the authors perform statistical tests of the calculated evaluation metrics to evaluate the significance of the performance difference across different methods? It is also better to incorporate another specialized method CelLink for comparative analysis (bioRxiv preprint doi: https://doi.org/10.1101/2024.11.08.622745)

2. For the ablation study, the performance improvements from individual components appear minor. Could the authors add statistical tests to substantiate these findings? Furthermore, to isolate the contribution of the hypergraph architecture more precisely, it would be valuable to include a comparison where only the hypergraph learning module is replaced with a general graph learning module, while the contrastive learning component is retained.

3. The paper highlights the significance of the hypergraph learning module within the MMIHCL framework; however, the corresponding description in the methodology section lacks sufficient detail. Specifically, the methodology section notes that W in Equation 6 is a dynamically learnable parameter matrix encoding high-order intrinsic relationships among the n embeddings of E. In contrast, conventional hypergraph learning typically entails the explicit construction of a hypergraph structure (e.g., each node and its K-nearest neighbors form a hyperedge), followed by the derivation of each node’s embedding via an information propagation mechanism that transfers information from nodes to hyperedges and subsequently back to nodes. Could the authors provide a rationale of such a design?

4. The paper briefly outlines the selection criteria for hyperparameters in the experimental setup. Could the authors perform a sensitivity analysis on key hyperparameters, such as the initial k-value employed to generate the adjacent cell graph and the regularization term coefficient λ in Equation 9? Furthermore, the number of iterative optimization rounds T in Stage 3 is set to a maximum of 3—can this be attributed to the sufficiently robust initialization achieved in Stage 2?

5. Are there new biologically findings compared to other methods?

Minor:

1. Line 104: surpassing MaxFuse and MMIHCL -> surpassing MaxFuse and scConfluence.

2. S6 Fig: The UMAP subgraphs for scConfluence—colored by data modality and cell type—should be adjusted to ensure consistency.

3. The method names on Fig. 3 and Fig. 4 are two big and please adjust it to match the font size of other texts.

Have the authors made all data and (if applicable) computational code underlying the findings in their manuscript fully available?

Reviewer #1: Yes

Reviewer #2: None

Reviewer #3: Yes

PLOS authors have the option to publish the peer review history of their article (what does this mean?). If published, this will include your full peer review and any attached files.

Do you want your identity to be public for this peer review? For information about this choice, including consent withdrawal, please see our Privacy Policy.

Reviewer #1: Yes: Zhiyuan Yuan

Reviewer #2: No

Reviewer #3: No

Figure resubmission:
---

## [Decision Letter · Decision Letter 1]

11 Mar 2026

PCOMPBIOL-D-25-01825R1

Single-cell data integration across weakly linked modalities

PLOS Computational Biology

Dear Dr. Zhou,

Thank you for submitting your manuscript to PLOS Computational Biology. After careful consideration, we feel that it has merit but does not fully meet PLOS Computational Biology's publication criteria as it currently stands. Therefore, we invite you to submit a revised version of the manuscript that addresses the points raised during the review process.

We look forward to receiving your revised manuscript.

Kind regards,

Jifan Shi

Academic Editor

PLOS Computational Biology

Ilya Ioshikhes

Section Editor

PLOS Computational Biology

Additional Editor Comments:

The authors have addressed most of the reviewers' comments. However, two questions from one reviewer remain unresolved. Therefore, we would appreciate it if the authors could respond to these remaining comments and submit a revised version for another round of review.

Journal Requirements:

Reviewers' comments:

Reviewer's Responses to Questions

Comments to the Authors:

Please note here if the review is uploaded as an attachment.

Reviewer #1: Good work.

Reviewer #2: Comments

The authors have addressed most of my concerns; however, two issues still require further clarification.

First, regarding my first-round Comment #4:

It is recommended to include additional ablation experiments involving the combined removal of core modules. Specifically, for the three key components (akNN, Hypergraph, and Filter), all pairwise combinations as well as the complete triple-module removal scenario should be designed as control settings. By quantifying the variation in core performance metrics under different module combinations, the following aspects should be explicitly clarified:

(1) Which module combinations exhibit significant synergistic gains;

(2) The specific contribution ratio of synergistic effects to the integration performance of weakly linked data;

(3) Whether module synergy serves as the core innovation that distinguishes MMIHCL from existing methods.

Such additional analyses would further strengthen the methodological rigor and the validity of the claimed contributions.

Second, regarding my first-round Comment #5:

Although the authors have included CelLink as a baseline, the comparison set still lacks coverage of other leading technical paradigms for weak-linkage integration published after 2023. To substantiate the claim of “optimal performance,” MMIHCL should be benchmarked against state-of-the-art methods from distinct methodological families.

Reviewer #3: The authors have well addressed my comments. However, to improve the clarity and readability of the figures, I suggest the authors enlarge the font size, as some of the text appears too small in its current form. For the installation instruction of the package, please also provide a clear command to install it.

Have the authors made all data and (if applicable) computational code underlying the findings in their manuscript fully available?

Reviewer #1: Yes

Reviewer #2: Yes

Reviewer #3: Yes

PLOS authors have the option to publish the peer review history of their article (what does this mean?). If published, this will include your full peer review and any attached files.

Do you want your identity to be public for this peer review? For information about this choice, including consent withdrawal, please see our Privacy Policy.

Reviewer #1: Yes: Zhiyuan Yuan

Reviewer #2: No

Reviewer #3: No

Figure resubmission:
---

## [Decision Letter · Decision Letter 2]

10 Apr 2026

Dear Dr. Zhou,

We are pleased to inform you that your manuscript 'Single-cell data integration across weakly linked modalities' has been provisionally accepted for publication in PLOS Computational Biology.

Best regards,

Jifan Shi

Academic Editor

PLOS Computational Biology

Ilya Ioshikhes

Section Editor

PLOS Computational Biology

Reviewer's Responses to Questions

Comments to the Authors:

Please note here if the review is uploaded as an attachment.

Reviewer #2: Authors have addressed all my concerns.

Have the authors made all data and (if applicable) computational code underlying the findings in their manuscript fully available?

Reviewer #2: Yes

PLOS authors have the option to publish the peer review history of their article (what does this mean?). If published, this will include your full peer review and any attached files.

Do you want your identity to be public for this peer review? For information about this choice, including consent withdrawal, please see our Privacy Policy.

Reviewer #2: No

---

## [Editor Report · Acceptance letter]

PCOMPBIOL-D-25-01825R2

Single-cell data integration across weakly linked modalities

Dear Dr Zhou,

I am pleased to inform you that your manuscript has been formally accepted for publication in PLOS Computational Biology. Your manuscript is now with our production department and you will be notified of the publication date in due course.

With kind regards,

Anita Estes
